# Rapid detection of social interactions is the result of domain general attentional processes

Jonathan C. Flavell[1]*, Harriet Over[1], Tim Vestner[2], Richard Cook[2], Steven P. Tipper[1]

1 Department of Psychology, University of York, York, North Yorkshire, United Kingdom, 2 Department of Psychology, Birkbeck, University of London, London, Greater London, United Kingdom

☯ These authors contributed equally to this work.

* jonathan.flavell@york.ac.uk

## Abstract

Using visual search displays of interacting and non-interacting pairs, it has been demonstrated that detection of social interactions is facilitated. For example, two people facing each other are found faster than two people with their backs turned: an effect that may reflect social binding. However, recent work has shown the same effects with non-social arrow stimuli, where towards facing arrows are detected faster than away facing arrows. This latter work suggests a primary mechanism is an attention orienting process driven by basic low-level direction cues. However, evidence for lower level attentional processes does not preclude a potential additional role of higher-level social processes. Therefore, in this series of experiments we test this idea further by directly comparing basic visual features that orient attention with representations of socially interacting individuals. Results confirm the potency of orienting of attention via low-level visual features in the detection of interacting objects. In contrast, there is little evidence for the representation of social interactions influencing initial search performance.

## Introduction

Encoding of viewed third party interactions appears to be automatic and can influence cognitive processes such as attention, working memory and longer-term memory [1]. Of particular note, the encoding of social interactions appears to be fast and automatic in that during a visual search task detection of two people facing towards each other is faster than for two people who ignore each other (see also [2]): a social priority effect (see Fig 1). It was argued that the detection of which people in a scene are interacting in this way is a critical early stage to quickly interpret the social information. According to this view, such social binding processes provide a basic visual input for more sophisticated processes, such as detection of deception, social affiliation, social dominance, and more generally a forward modelling ability to predict potential future actions.

**Data Availability Statement:** Data, stimuli, statistical models, and supplementary documents are available at osf.io/qxk8z.

**Funding:** This research was supported by a Leverhulme Trust grant awarded to SPT and HO

(RPG-2017-068), and by a European Research Council Starting Grant awarded to RC (ERC-STG-715824). The funders had no role in study design, data collection and analysis, decision to publish, or preparation of the manuscript.

**Competing interests:** The authors have declared that no competing interests exist.

**Fig 1. Social priority effect in the visual search array employed from Experiment 1 of Vestner et al. [1].** Detection of the target in the top-left location is faster when the pair are facing (Panel A) than when back-to-back (panel B).

Indeed, in our initial research programme investigating the form of representation mediating these social binding effects, we provided evidence for the role of representations of social interactions rather than low-level perceptual features driving the behaviour. This evidence was harvested from a series of studies examining visual search, working memory and longer-term memory. In that paper we determined that "...*results are consistent with the social binding hypothesis, and alternative explanations based on low level perceptual features and attentional effects are ruled out. We conclude that automatic midlevel grouping processes bind individuals into groups on the basis of their perceived interaction*" [1], pp 1251). More broadly, it is now well established that visual search processes are influenced by prior learning of the emotional properties of a stimulus, and such attention capture effects cannot be explained by low-level physical properties of a stimulus. Such studies have examined a wide range of situations from electric shock (e.g. [3]) to negative social feedback (e.g. [4], and [5] for review).

A central feature of such social interactions is of course joint attention via gaze direction (see [6] for review). That is, when two people are interacting they are typically attending to one another. Hence it is possible that attention processes evoked by the direction of gaze are contributing to the assessment of whether or not they are interacting, and consequently to the social binding priority effect. However, a critical issue concerns whether general attention orienting mechanisms are necessary and sufficient to account for the social binding priority effect ([7,8], see also [9] for review) or whether processes involving the representation of social agents is critical (e.g. [1,10]).

One approach to this issue is to examine effects with stimuli that do not have the mental states that mediate social interactions but nevertheless orient attention. Arrows have been shown to possess these properties. For example, arrows produce attention orienting effects that are very similar to the social attention cues of a person's gaze direction (e.g. [11]). Clearly, arrows are not biological stimuli with social intent and do not act in an interactive manner. Rather, arrow-like stimuli have intrinsic low-level physical properties that imply direction. The basic visual properties of direction can be seen, for example, in the shape of the arrow launched from a bow, the stream-lined shape of the fighter jet or sports car, or the body shape of the diving gannet. In each case the pointing shape exists to facilitate movement in a particular direction. Hence simple visual features, such as those possessed by arrows, imply direction.

If attention orienting by simple physical cues (independent from implied social interactions) is central to the previously observed effects then a clear prediction is that towards facing arrows will be detected faster than away facing arrows. Indeed, recent work by Vestner et al. [12] has demonstrated that this is the case. That is, just as towards facing people are detected faster than those facing away from one another (see Fig 1), so also towards facing arrows are detected faster than away facing arrows. That such simple stimuli with non-social properties

can produce the same effects challenges prior claims that they play no role in the effects observed by Vestner et al. [1].

However, though similar effects are obtained with low-level visual stimuli not containing social information, this does <u>not</u> demonstrate that the latter higher-level processes play no role in social binding processes. This is especially the case if effects are at ceiling. The brain represents multiple properties of the visual world, from lower-level features such as simple shape, colour, proximity and motion (e.g. [13,14]) to higher-level representations of object identity, emotion and social properties. Indeed, Ristic et al. [15] found that gaze and arrow cues are managed by separate systems to the same functional outcome. The parallel co-existence of multiple forms of representation across cortical and subcortical networks leads to the possibility that the effects of some internal representations, not observable in some tasks, might be detected in other situations. Therefore further work employing converging techniques is necessary. In the current series of experiments we further investigate the low-level visual features that influence visual search, but also simultaneously manipulate higher-level representations of third-party social interactions that can be congruent or incongruent with low-level properties. Such an approach avoids the interpretational issues caused by potential floor or ceiling effects.

## Our approach

In our new studies we manipulate these two forms of processing within the same experimental stimuli. Consider Fig 2, which shows examples of the displays employed in Experiments 1 (A&B) and 2 (C&D). These teardrop stimuli have low-level basic features that imply direction [16]. That is, in a basic Posner cueing task targets presented to the pointed side of the teardrop stimulus (Fig 2A) are detected significantly faster than targets presented to the round side of the stimulus (Fig 2B, and see Experiment 1). Hence in the visual search task featuring such stimuli, target detection will be faster when stimuli point towards (Fig 2C) rather than away (Fig 2D, and see Experiment 2) due to attention being jointly focused to one location, replicating Vestner et al. [12] with these new stimuli.

However, prior to the search task we employ a social learning stage where participants are presented with Heider & Simmel [17] type video displays, adapted from those of Over & Carpenter [18]. Hereafter these videos are described as 'social priming videos'. In such displays,

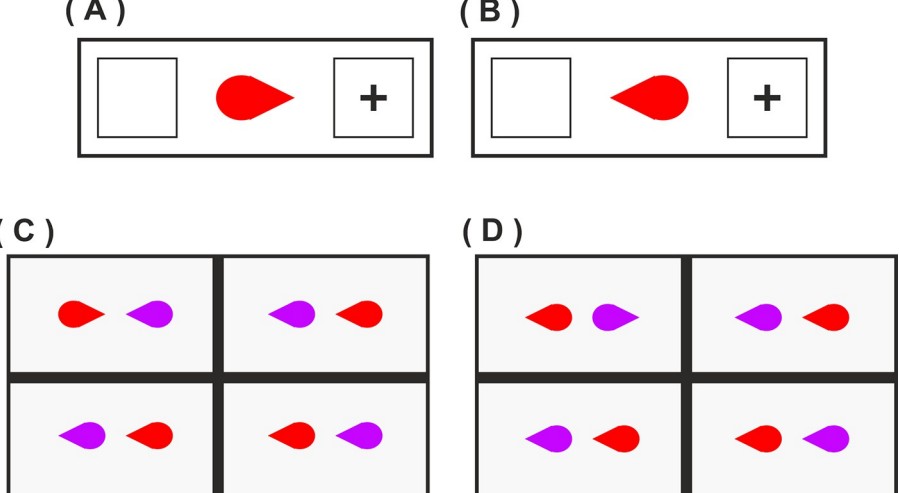

**Fig 2.** Examples of Posner cueing (A&B, Experiment 1) and visual search (C&D, Experiment 2, target pair in the top left quadrants) displays featuring stimuli with directional cues.

stimuli move in particular spatial patterns that evoke a powerful experience of complex social interactions, such as inclusion/exclusion, teasing, and cooperation/competition, and even characters with particular personalities. This is a potent technique which appears to evoke universals in social perception based on object interactions that are similar across cultures [19], develop early [18,20,21] and can reveal individual differences in social perception (e.g., in autism see [22]). Furthermore, observation of such displays showing positive or negative social interactions can influence basic perceptual processes such as judgments of distance (e.g. [23]) and they can prime particular states that can influence behaviour at a later time (e.g. [18,24,25]).

In these displays, we manipulate the orientation and direction of motion of the socially interacting objects. For example, in one condition during the pro- and anti-social interactions of the objects, the pointing end is equivalent to the face/front end. However, in a second condition the stimuli are reversed such that the round end is equivalent to the face/front end during the social interactions.

A critical features of our displays is the direction of motion. Hernik et al [24] have shown that motion direction can reliably disambiguate the front acting part of an object from it's back. Furthermore, classic studies from ethology show that direction of motion can influence the identity of an ambiguous stimulus. For example, the famous hawk/goose stimulus is perceived as a hawk when the short end is the movement direction, and as a goose when the long end is the motion direction. This direction of motion of the goose/hawk silhouette can determine whether predator avoidance responses are evoked in a number of bird species [26]. A second important feature in our video displays beyond motion direction is the interaction with other objects. The side of a stimulus that interacts with and has an influence on other objects is perceived to be the active action end of the object (e.g. [24]). A final aspect of our motion stimuli is that they contain no intrinsic visual features that could be construed as a face. Previous work has shown that features to one side of an object can introduce directionality to an object representation, possibly evoked by attention capture by the features (e.g. [27]). Hence to avoid this potential confound we have employed very simple objects with no salient intrinsic face-like features. Via this approach we attempt to manipulate the interpretation of exactly the same stimuli via prior social priming. Fig 3 provides a schematic example of the motion displays where the round end is perceived as the active "face" end, but we recommend viewing the actual videos at osf.io/qxk8z.

Because observation of such third-party interactive displays evokes a powerful sense of agency, personality, and the achievement of social goals [18,21,28] they should create internal representations of individuals that socially interact. Certainly, looking ahead, our displays are sufficiently potent to evoke a face/front end of targets in the overwhelming majority of participants. With this in mind, the search performance of the participants who have observed the round end of the interacting objects as the "face" might be influenced in a subsequent visual search task. There are three data patterns that identify the roles of the two processes of low-level visual direction features and higher-level social interaction:

First, if only low-level stimulus feature-based attention processes are at play, then independent of what participants observe in the prior social priming videos, detection of targets in Fig 2C (point inwards) will always be faster than the targets of Fig 2D (round inwards).

Second, internal representations of socially interacting individuals might dominate low-level visual features. If so, search performance will be driven by the participant's experience of the prior social priming videos. Detection of targets in Fig 2C (point inwards) would be faster than Fig 2D (round inwards) when the pointed end of the objects are represented as "face"; whereas the opposite pattern will be observed in participants who observe the round end as "face" in the social priming video.

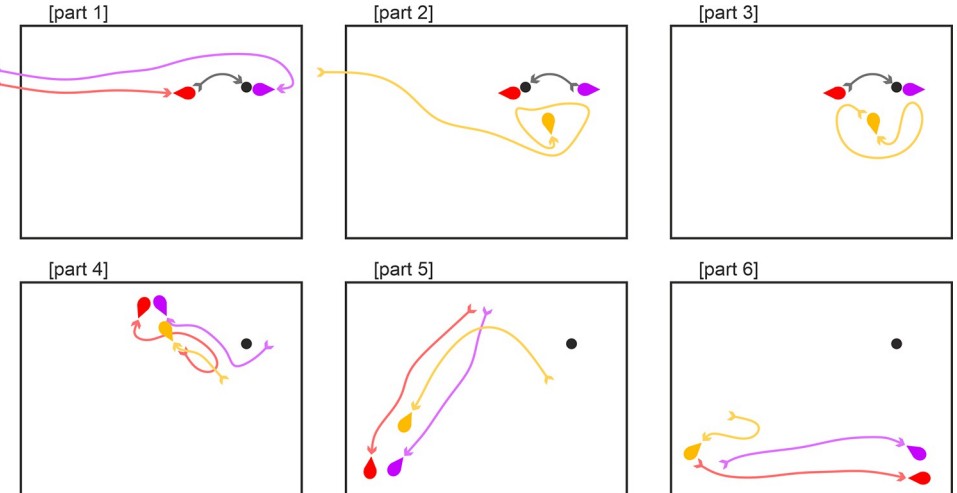

**Fig 3. Schematic representations of the experiment characters interacting in the priming video of Experiments 2, 3 and 5.** In this example, the round end of the teardrop is the leading edge of the motion direction and perceived as the face. The arrow headed lines were not present in the video and are shown here only to illustrate movement. Note that the characters (shown here as red, purple and yellow teardrop shapes) differed in Experiment 5 (see appropriate section). Part 1: Purple and Red enter the scene, approach the black ball then toss it back and forth. Part 2: Yellow enters the scene, approaches Purple and Red but receives no response. Part 3: Purple and Red toss the ball back and forth then Yellow reproaches each of them. Part 4: Purple and Red move away then Yellow pursues. Part 5: Again, Purple and Red move away then Yellow pursues. Part 6: Purple and Red move away, yellow considers following but then slowly moves away. Full video at osf.io/qxk8z.

Finally, it is possible that both the low-level visual feature processes and higher-level social interaction processes will influence search simultaneously. In this case, there will be an interaction between the social priming condition and the point towards (Fig 2C) vs point away (Fig 2D) search effect where the largest effect will be observed after priming of social interactions with the pointed end as "face" due to the combined facilitation effects of low-level and higher-level stimulus properties. In contrast, after social priming where the round end is "face" there will be competition between low-level properties of pointedness and higher-level social representations, reducing the point towards vs away effect.

Because this is a somewhat complex article containing 7 experiments utilising different techniques to examine the roles of both low-level visual features and higher-level social representations, we felt it would facilitate comprehension to preview our findings of each experiment at this point (see Fig 4). We found that attention orienting visual features (Experiment 1 and 4) dominated visual search for target pairs even following social priming (Experiment 2) and social priming with semantic labelling and biological animacy (Experiments 3 and 5). Further, even when using targets which lacked low-level attention orienting shape features (Experiment 6) social priming with semantic labelling and biological animacy was insufficient to orient attention in a visual search task (Experiment 7). With regard to rapid visual search for interacting pairs, this consistent pattern of findings provides evidence for attention orienting by visual features, and no evidence for attention orienting by social representation.

## Experiment 1 ('Teardrop' attention pre-test)

Because the teardrop stimuli to be presented in the social priming video are novel and have not been used before, we have to first demonstrate their ability to automatically orient attention. That is, unlike gaze (e.g. [6]) and arrows (e.g. [11]), which have been extensively studied and produce robust automatic shifts of attention, it is possible that the teardrop stimuli we

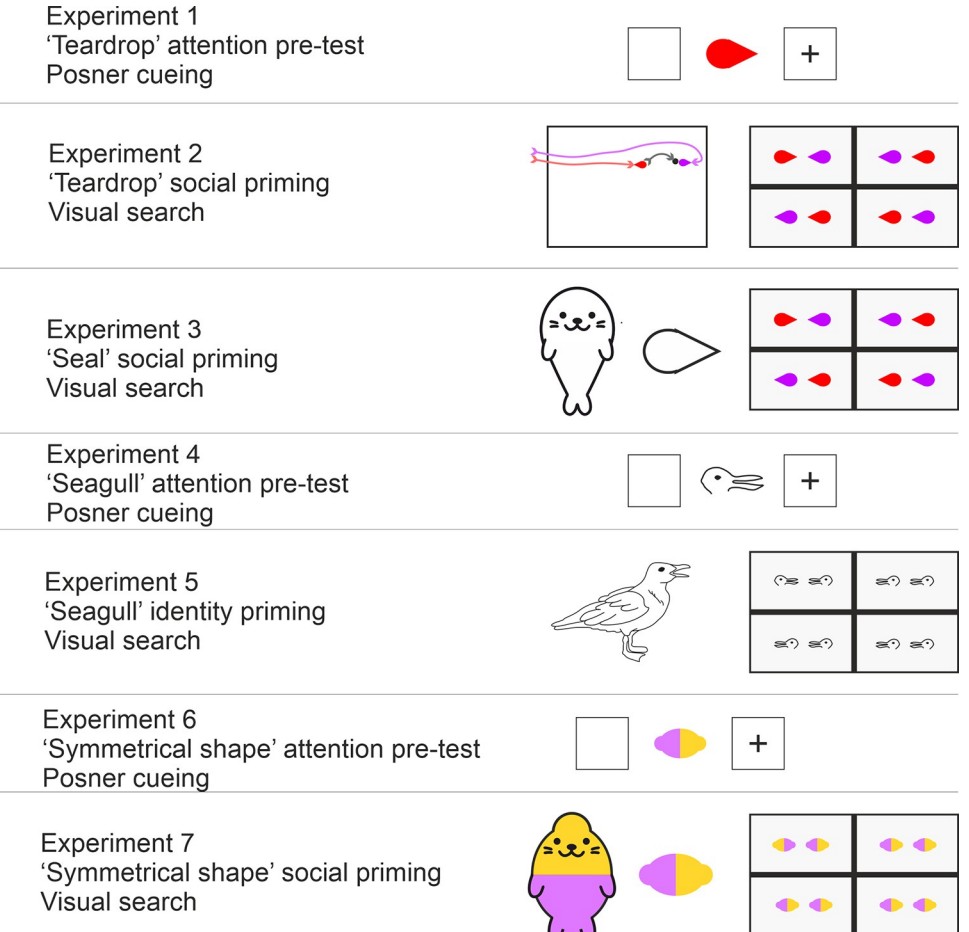

**Fig 4. Representations of each experiment with associated stimuli.**

employ will not have these basic low-level orienting properties. Therefore in Experiment 1, we present the teardrop in a Posner cueing design. The teardrop stimulus is presented in the centre of the screen and a target is presented either to its left and right. Because this cue is irrelevant to the participant's task, and it does not predict target location, we can examine whether it evokes automatic attention orienting responses. We predict that the basic visual property of the pointed end of the object will shift attention to that side of space, just as gaze and arrow cues do.

## Method

**Apparatus.** The experiment was built and hosted in Gorilla Experiment Builder (www. gorilla.sc, [29]). Browsers were restricted to Chrome, Firefox, Safari, Edge and Internet Explorer. Devices were restricted to desktop and laptop computers.

**Design.** Participants completed a practice block and a task block. Before each block, participants were shown instructions on the screen. Verbatim copies of the instructions given to participants are available at osf.io/qxk8z.

At the start of a trial, two identical boxes appeared to the left and right of screen centre. After 500ms the cue would appear in the centre and then either 200 or 600 ms later a target (a cross) or a distractor (a circle) would appear in either the left or right box. If the target cross

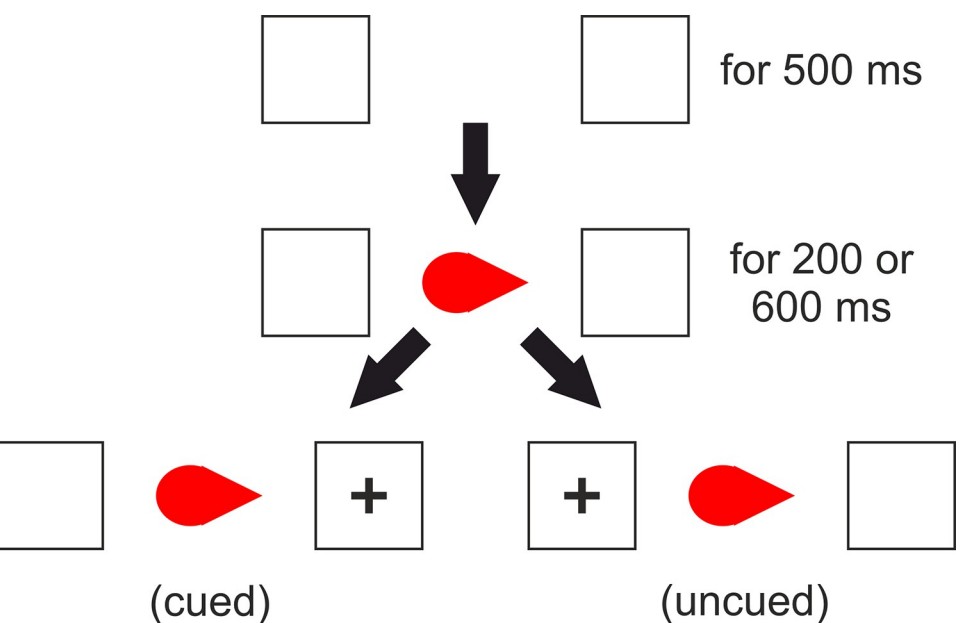

**Fig 5. Schematic representation of a trial in Experiment 1.**

appeared then participants were to press 'F' if it was on the left or 'J' if it was on the right. However, if the distractor circle appeared then participants were to not press anything and should simply wait for the end of the trial. Participants were not instructed on which fingers to use for the task though in pilot testing all participants used their left index finger for 'F' (left side of the keyboard) and right index finger for 'J' (right side of the keyboard) on a QWERTY keyboard. If an incorrect response was made (wrong side reported for a cross or any response to a circle) then a 'thumbs down' appeared over the cue for 500 ms before the trial ended. Target trials ended when a response was made. Distractor trials ended when either a response was made or 2000 after distractor appearance. At the end of a trial the screen was blank. See Fig 5.

Cues in the practice block were black dots (lacking directional attention cue) whereas cues in the task block were teardrop shapes (i.e. possessing directional attentional cue, see Fig 5) in purple or red. Cued trials are those in which the cue pointed to the same side that the target/distractor would appear, and uncued trials are those in which the cue pointed to the opposite side that the target/distractor would appear. The point direction is defined by the acute end of the cue (i.e. as a standard arrow). The target circle had the same height and width dimensions as the target cross. All stimuli and stimulus size details are available at osf.io/qxk8z.

The practice block had 12 trials. Half had a stimulus onset asynchrony (SOA) of 200 ms and half had a SOA of 600 ms. Of each set of 6 there were 2 leftwards and 2 rightwards target trials, and 1 leftwards and 1 rightwards distractor trial. The task block contained 112 trials. Half had a SOA of 200 ms and half had a SOA of 600 ms. Of each set of 56 there were 12 cued leftwards target trials, 12 uncued leftwards target trials, 12 cued rightwards target trials, 12 uncued rightwards target trials, 2 cued leftwards distractor trials, 2 uncued leftwards distractor trials, 2 cued rightwards distractor trials, and 2 uncued rightwards distractor trials.

**Participants & analysis.** Protocols were approved by the University of York's Psychology Departmental Ethics Committee and were in accord with the tenets of the Declaration of Helsinki. Participants were recruited through Prolific with filters of age from 18 to 50 and vision as normal or corrected to normal. Thirty-two participants were tested but 26 remained following exclusions (see Data exclusion and analysis). No participant completed more than one

experiment in this programme of experiments. Informed consent was obtained prior to participation. Data was collected on the 12th and 13th of December 2019.

Bayesian analysis was planned for all experiments in this manuscript using JASP v0.13 [30]. Participant N thresholds are not necessary to interpret Bayes Factors (BFs) which indicate the weight of evidence in favour of, or against, the null/alternative hypothesis (specified in each model). As such, power analyses were not performed and our sample size of 26 was based on typical simple attention cueing and visual search designs. We report evidence categories [31] after each BF to aid interpretation of values e.g. ($BF_{10}$ = 3.005e+7 [*extreme evidence for H1*],...).

**Data exclusion.** Full exclusion details can be found at osf.io/qxk8z. Briefly: two participants were excluded due to error rate (>25% errors in either the target (i.e. >24) or distractor (i.e. >3) trials); and 4 participants were excluded due to too few remaining trials following RT exclusion (<75% in any SOA × cue condition). The mean ± SD percentage of trials remaining in each SOA × cue for each participant 94.4 ± 6.2%.

## Results & discussion

Reaction times are shown in Fig 6. Bayesian repeated measures ANOVA on RTs with within subject factors of SOA (200/600 ms) and Cue (cued/uncued) support a model including both main terms ($BF_{10}$ = 3.005e+7 [*extreme evidence for H1*], $p$(H1|Data) = .603; SOA $BF_{incl.}$ = 1.376e+7; Cue $BF_{incl.}$ = 4.493). Reaction times were shorter for the 600 ms SOA than for the 200 ms SOA, and were shorter when the target was cued (point towards the target) than when the target was uncued (point away from the target). Frequentist modelling supports these findings. Models at osf.io/qxk8z.

The results are clear and demonstrate that these teardrop stimuli do indeed produce automatic attention orienting. That is, even though the cue is non-predictive of target location and to be ignored, it shifts attention rapidly (within 200ms) and the attention shift remains stable

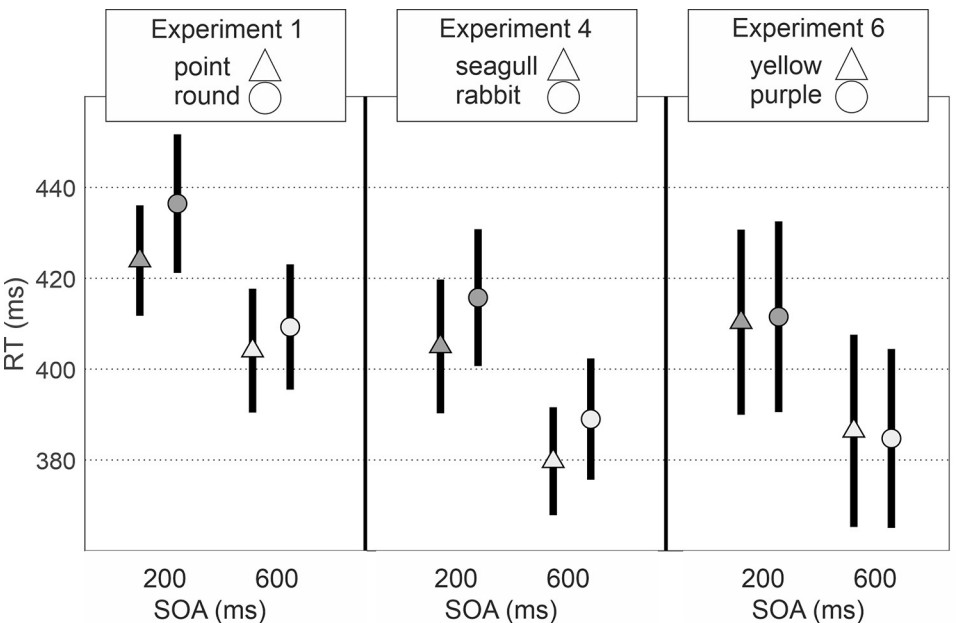

**Fig 6. Mean (±95% confidence interval) reaction times to target appearance side in each SOA × cue condition for Experiments 1, 4 and 6.**

until at least 600ms. Therefore, after featuring in social priming videos, these teardrop stimuli will have the basic low-level visual properties that evoke attention shifts and higher-level priming of third-party interactions.

## Experiment 2 ('Teardrop' social priming)

This experiment attempts to manipulate the properties of the teardrop objects from Experiment 1 by featuring those objects in social priming videos (adapted from Over & Carpenter [18]) in which they are observed interacting in dynamic and complex spatial movement patterns. These patterns of motion evoke a salient and powerful impression of third-party social interactions, which display complex social states such as cooperation, in- vs out-group structure, and finally rejection and sadness.

### Method

**Apparatus.** Participants sat at a table in a dimmed room facing a 23" touch screen monitor (Iiyama (Tokyo, Japan) ProLite T2735MSC-B2, 1920×1080 pixels) at approximately 50 cm distance. A keyboard was positioned on the table between the participant and the screen. Participants and the keyboard spacebar were positioned at the screen's horizontal centre. Stimulus presentation (60Hz) and response recording were achieved using custom scripts and Psychtoolbox 3.0.11 [32–34] operating within Matlab 2018a (The MathWorks Inc., Natick, USA) on a PC (Dell (Round Rock, USA) XPS, Intel (R) Core (TM) i5-4430, 3 GHz CPU, 12 GB RAM, 64 bit Windows 10 Enterprise).

**Experiment design.** Participants completed a practice block, then four task blocks, and finally a target orientation question. In the practice and task blocks participants searched for and then touch the target pair in a set of four simultaneously presented pairs. Participants were shown the target pair on an instruction screen before each of those blocks. The target pair changed between blocks and was not present on every trial. The target orientation question was to check that our priming videos had effectively demonstrated the face (agency/attention) direction of targets.

**Practice & task trial composition.** At the start of a trial a cross divided the screen into four sections indicating that the participant should press and hold the space bar (Fig 7A). Pressing the spacebar caused a pair of characters to appear in each section (Fig 7B). Participants searched for the target pair and, upon discovery, released the space bar and reached out

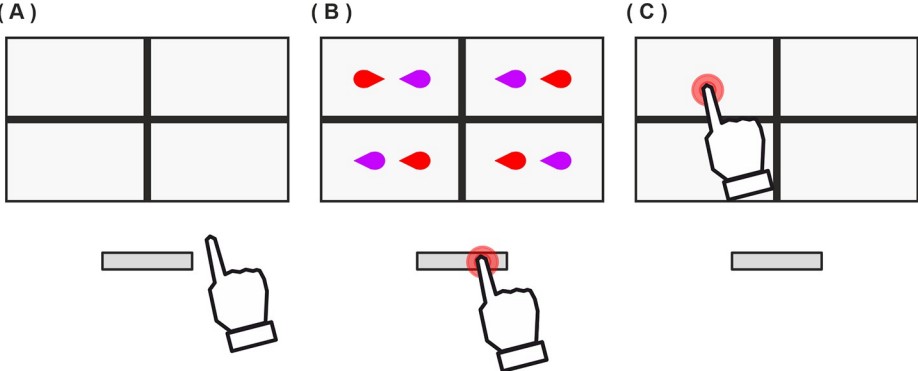

**Fig 7.** A-C) Schematic representation of a trial in Experiments 2, 3, 5 and 7. In this example the target pair is that with points together (upper left quadrant). Note that the targets (shown here as red and purple teardrop shapes) differed in Experiment 5 (see appropriate section). Target dimensions were ~30×15 mm and the distance between targets in a pair was ~15 mm.

to tap that pair with the same finger that was pressing the spacebar (Fig 7C). Releasing the space bar caused all four pairs to disappear so the target had to be identified before beginning a reaching action. If the target pair was not present (i.e. all four pairs were distractors) then the participant should keep holding the space bar down until the end of the trial. A trial ended either when the screen was tapped or 5s after the space bar was first pressed. Reaction times were measured from the moment the four simultaneously presented pairs appeared to the moment of space bar release. Movement time is measured from the moment of space bar release to the moment of screen contact.

**Target and distractor pairs.**   Each of the four pairs presented on a trial was made-up of two shapes that possessed directional attentional cues. The shapes in the target pair either cued inwards or outwards whereas the shapes in the other pairs (the distractor pairs) always cued both leftwards or both rights (see Fig 7B). All stimuli and stimulus size details are available at osf.io/qxk8z.

**Practice block.**   The practice block was to familiarise participants with the task. Participants were instructed to "find and tap the target pair as quickly as possible" but if the target pair was not present (i.e. only distractor pairs were present) then they were to hold down the space bar until the end of the trial. Practice block instructions were presented on-screen and verbally by the experimenter. Participants had the opportunity to ask questions before starting the task blocks which they would complete in isolation. Verbatim copies of the instructions given to participants are available at osf.io/qxk8z.

**Task blocks and social priming.**   Participants were shown a video prime before each task block. This 55s Heider & Simmel (1944) type video was adapted from those of Over & Carpenter (2009). It featured the purple and red teardrop shapes from Experiment 1 as well as a yellow teardrop shape. The purple and red shapes were seen playing and acting jointly to exclude the yellow shape, which was trying to play with them (see Fig 3, and osf.io/qxk8z for full video). The intent was to prime participants to understand that either the pointed or rounded end (see Conditions) of the shapes was their "face". The pointed and rounded videos were identical apart from the orientation of the teardrop which was mirrored to create a pointed or rounded prime. The prime for pointed or round alternated between participants.

**Conditions.**   Every block contained standard trials (in which one of the four pairs was the target pair) and catch trials (in which all four pairs were distractor pairs). The orientation of target pair (pointing inwards or outwards) for the practice block was determined randomly between participants. The orientation of the target pair for the 4 task blocks was alternated in an A-B-A-B or B-A-B-A pattern as participants completed the experiment. For example, participant 1 would receive the target pairs [inwards-outwards-inwards-outwards] across blocks then participants 2 would receive the target pairs [outwards-inwards-outwards-inwards] across blocks (see Fig 8).

On a catch trial, pairs would all point leftwards or all point rightwards. On a standard trial, one pair would be in the target orientation and the other pairs would all point leftwards or all point rightwards. Within each pair the targets would be in the colours they had seen in the social priming video. Two left hand targets from different pairs would be one colour and two would be the other colour. All colours and distractor pair orientations were counterbalanced within task blocks. The section of the screen in which the target would appear was randomised between trials.

The practice block contained 6 trials. There were two catch trials. In one catch trial all targets pointed leftwards and in the other all targets pointed rightwards. For the remaining four trials the orientation of the targets (shown in the instructions) was determined randomly between participants. The task blocks each contained 24 trials. There were four catch trials. In two catch trials all targets pointed leftwards and in the others all targets pointed rightwards.

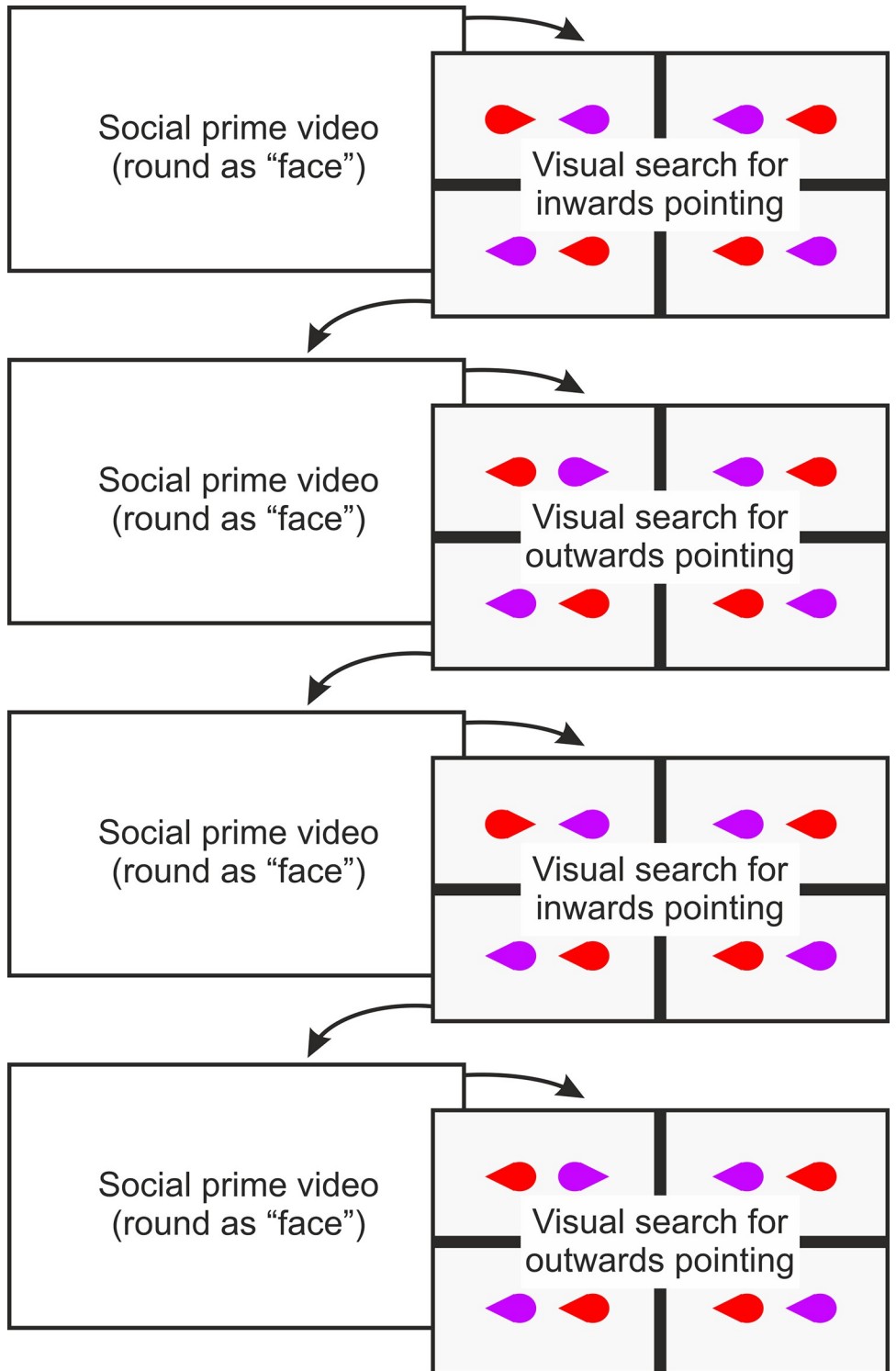

**Fig 8. Schematic of experiment design demonstrating the progress of a participant in the round "face" social prime condition with inwards-outwards-inwards-outwards visual search blocking.**

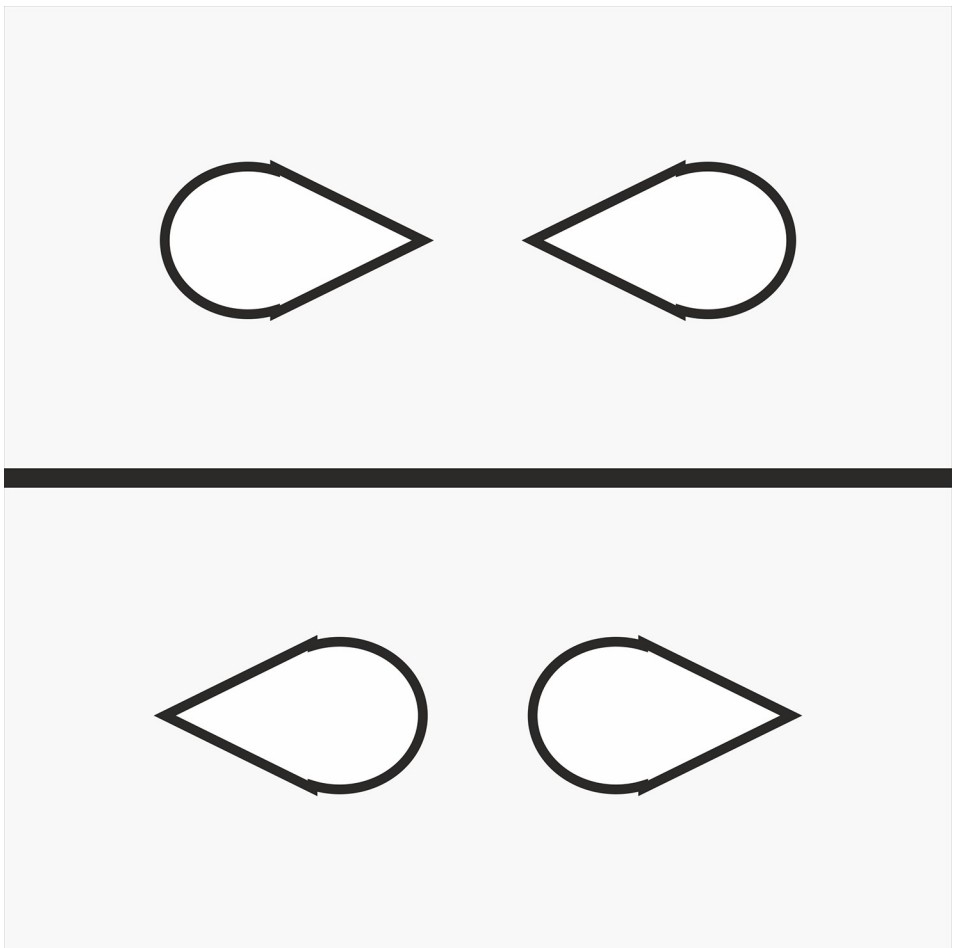

**Fig 9. Schematic of the target pair presentations for the target orientation question.**

For the remaining twenty trials the orientation of the targets was determined by block number. Half of these trials had distractors pointing leftwards and half pointing rightwards.

**Target orientation question.** After completing the final task block participants were presented with a single question about target orientation. Participants were presented with two pairs of targets in the centre of the screen (see Fig 9) and asked to tap on the pair that was facing each other. The targets were the same size and shape as those in the practice block but were white rather than red or purple.

**Participants.** Protocols were approved by the University of York's Psychology Departmental Ethics Committee and were in accord with the tenets of the Declaration of Helsinki. Participants were recruited through the University of York's Psychology Department participant recruitment system. Informed consent was obtained prior to participation. For the pointy social prime 33 participants were tested and 26 (age mean±SD = 21.7±9.1, 6 male, 1 undisclosed) remained following exclusions. For the round social prime 31 participants were tested and 26 (age mean±SD = 19.8±2.4, 4 male) remained following exclusions.

Note—the data in this experiment (and the other social/identity priming experiments: Experiments 2, 3, 5, 7) are from participants recruited and tested 'in-lab' whereas the data from pre-test experiments (Experiments 1, 4, and 6) are from participants recruited and tested 'on-line'.

**Data exclusion & analysis.** Full exclusion details can be found at osf.io/qxk8z. Briefly: 10 participants were excluded due to error rate (errors on >20% of trials); 1 participant was excluded for using two hands; and 1 participant was excluded due to too few remaining trials following reaction time and movement time exclusion (<75% trials). The mean ± SD percentage of trials remaining in each condition for each participant was 98.4±3.2 in the pointy face prime condition and 97.4±4.0 in the round face prime condition. Following all exclusion there were an equal number (n = 13) of participants in the A-B-A-B and B-A-B-A designs of each prime condition. Movement time is not considered a principle indicator though analysis of movement time is provided for all visual search experiments (Experiments 2, 3, 5 and 7) at osf. io/qxk8z for completeness.

## Results & discussion

Bayesian repeated measures ANOVA on RTs (see Fig 10) with a within-subjects factor of target orientation (point inwards/point outwards) and a between-subjects factor of social prime type (pointed face/rounded face) support a model including only the target orientation term ($BF_{10}$ = 391375.418 [*extreme evidence for H1*], $p$(H1|Data) = .697; $BF_{incl.}$ = 293804.334). Reaction times were faster when detecting inwards pointing targets regardless of the prime.

In the target orientation task (at the end of the experiment where participants were required to report which way the objects faced) Bayesian binomial tests (test value = 0.5) indicated that the majority of participants were able to identify the facing pair in both the rounded (24/26 participants, $BF_{+0}$ = 15295.424 [*extreme evidence*]) and pointed prime conditions (19/26 participants, $BF_{+0}$ = 7.485 [*moderate evidence*]).

Frequentist modelling supports all findings. All models at osf.io/qxk8z.

This experiment has demonstrated quite clearly that 1) our priming technique were effective, and 2) that low-level properties of attentional orienting are computed and can guide visual search. That is, just as with towards facing faces or arrows, we find the attention orienting direction of teardrop stimuli facilitates target detection. Importantly, we find the prior exposure to the video displays did not influence the subsequent visual search performance. That is, whether participants observed social interactions where the pointed end of the objects

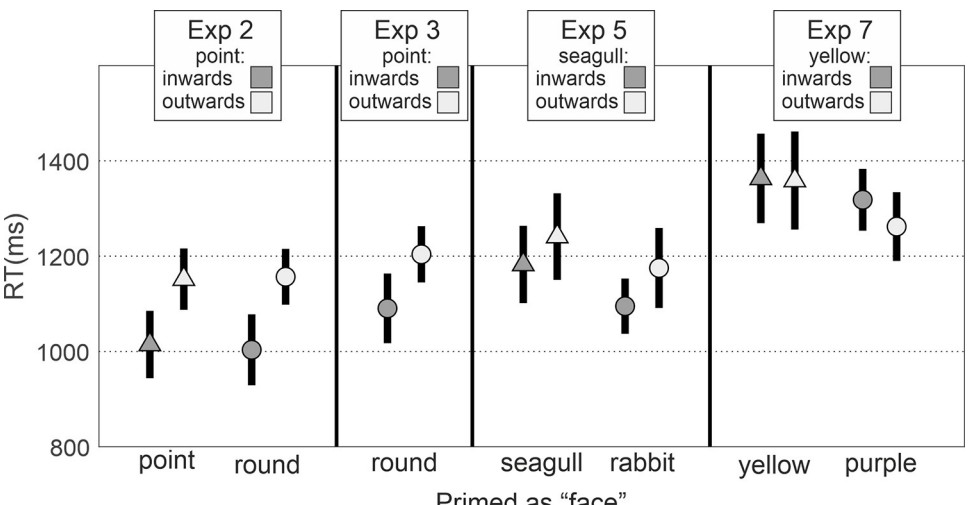

**Fig 10. Mean (±95% confidence interval) reaction times to target pairs in the visual search task for each target orientation × social prime condition in Experiments 2, 3, 5 and 7.**

were perceived as the "face" or the round end as the "face", had no effect on target search performance.

However, clearly, to propose that high-level third-party social interaction relationships play no role in the visual search performance would be premature based on one experiment. It might be the case that although the video priming technique appears to have influenced object perception when tested at the end of the experiment, it is possible that during visual search such representations are less salient. Therefore Experiment 3 is a replication, but with a more compelling approach to boost the priming effects of the social interaction videos. Previous research has clearly demonstrated attention cueing effects with cartoon characters, avatars [35] and animals [36]. Hence we decided to increase the potency of object identities by first introducing the teardrops as seals and show some initial interactions before showing the social priming video and completing the search task of the present experiment. Providing semantic labels to the objects might further increase the sense of biological animacy when observing the third-party interactions.

## Experiment 3 ('Seal' social priming)

### Method

**Apparatus.** The apparatus was identical to that of Experiment 2.

**Design.** The design was identical to that of Experiment 2 with two exceptions. First, just prior to viewing the social priming video a further priming block provided the stimuli with the semantic identity–the teardrop shapes were described as seals. Second, participants were never cued to view the pointed end as the "face" of the teardrop shape.

The new priming block was intended to give a much stronger cue to view the rounded end of the teardrop as the "face". At the start of the priming block, participants were told that this experiment was about seals. They were shown a modified target shape as a cue (Fig 11A) and then shown what the seals would look like in this experiment (Fig 11B). Next they were shown three videos in which a lone seal moved (Fig 11C), two seals greeted each other (Fig 11D), and three seals were dyed from white to red, yellow and purple (Fig 11E). This last video was to colour the white target shapes to match those of Experiment 2 thus allowing an identical task block presentation. Immediately after this further semantic priming, participants observed the same social priming video of Experiment 2, and these latter videos were then observed before every visual search block (Fig 3). All stimuli and stimulus size details are available at osf.io/qxk8z.

**Conditions.** Conditions were identical to Experiment 2 but participants never saw the pointed front prime version of the social priming video illustrated in Fig 3 i.e. only the round end (congruent with the seal head) was ever primed as the "face".

**Participants.** Protocols were approved by the University of York's Psychology Departmental Ethics Committee and were in accord with the tenets of the Declaration of Helsinki. Participants were recruited through the University of York's Psychology Department participant recruitment system. Informed consent was obtained prior to participation. Twenty-nine participants were tested but 26 (age mean±SD = 19.6±3.5, 2 male) remained following exclusions (see Data exclusion and analysis).

**Data exclusion and analysis.** Full exclusion details can be found at osf.io/qxk8z. Briefly: two participants were excluded due to error rate (errors on >20% of trials); and 1 participant was excluded due to few remaining trials following RT and MT exclusion (<75% trials). The mean ± SD percentage of trials remaining in each condition for each participant was 94.8 ±11.5. Following all exclusion there were an equal number (n = 13) of participants in the A-B-A-B and B-A-B-A designs.

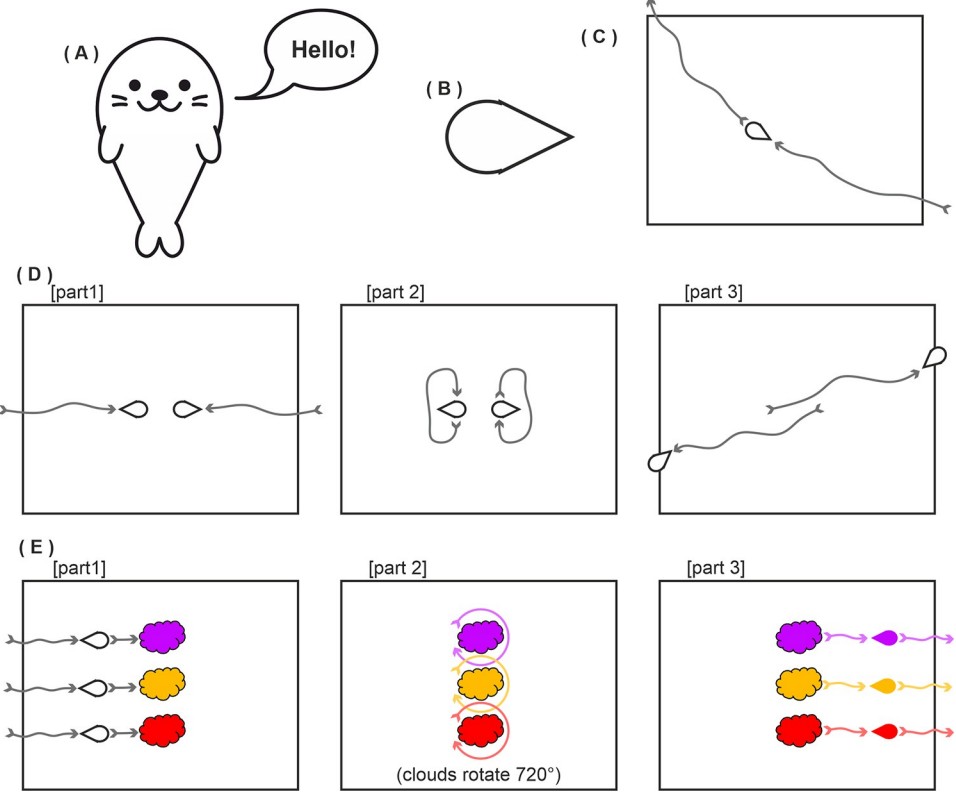

**Fig 11. Stimuli and primes in Experiment 3.** A) Seal prime. B) Seal target. C) Schematic representation of the first video prime in which a seal moves right to left. D) Schematic representation of the second video prime in which two seals approaching each other (part 1), greet each other (part 2) and then moving on (part 3). D) Schematic representation of the third video prime in which three seals move into coloured clouds (part 1), the clouds rotate 720˚ (part 2), and the seals leave the clouds having taken on colour (part 3). All stimuli available at osf.io/qxk8z.

## Results & discussion

Bayesian repeated measures ANOVA on RTs (see Fig 10) with a within-subjects factor of target orientation (point inwards/point outwards) support a model including the target orientation term ($BF_{10}$ = 10.866 [*strong evidence for H1*], $p$(H1|Data) = .916). To confirm the consistency of these effects, a further analysis combining the results of Experiment 3 with those of Experiment 2 did not favour a model featuring social prime type (pointed face/rounded face/rounded (seal) face). Instead, it supported a model featuring only the target orientation term ($BF_{10}$ = 1.162e+7 [*extreme evidence for H1*], $p$(H1|Data) = .631). Reaction times were faster when detecting inwards pointing.

In the target orientation task, Bayesian binomial tests (test value = 0.5) indicated that participants were able to identify the facing pair (which were primed by the video interaction) (26/26 participants, $BF_{+0}$ = 4.971e+6, [*extreme evidence*]).

Frequentist modelling supports all findings. All models at osf.io/qxk8z.

Again, the results are clear. The priming was effective at communicating target front and, despite semantic labelling and multiple observations of the round end as the "face" during social priming, the low-level physical property of the pointedness automatically triggered attention orienting which resulted in faster detection of targets pointing inwards.

## Experiment 4 ('Seagull' attention pre-test)

When arguing for the absence of an effect, it is important to employ alternative approaches. Such converging techniques increase our confidence in the conclusions drawn from the

observed data. Therefore, in the next two experiments we use an ambiguous figure for which two sides can be viewed as its "face". Such ambiguous figures have intrigued students of perception for many years and here we use a version of Fisher's [37] well-known ambiguous rabbit/duck figure (Fig 12A). Prior to running Experiment 4, we explored whether this version of the duck/rabbit was in fact seen as those two animals. Anecdotal reporting from naïve viewers indicated the rabbit was always perceived but that the alternative interpretation was more strongly reminiscent of a seagull than a duck. Consequently, our images were described as *rabbit* or *seagull* to participants and for the remainder of this manuscript.

Our intention in this experiment is to bias the interpretation of this stimulus' "face" using priming techniques and then show the stimuli in a visual search task in which the stimuli are "facing" towards or away from one another, as in the previous visual search tasks. Our logic is the same as Tinbergen's [26] goose/hawk ambiguous figure where the same physical object can have quite different identities.

In the present experiment we explore the attention cueing nature of our rabbit/seagull stimulus using the Posner design of Experiment 1. Because the potential attention cueing properties of these stimuli have never been examined, it was necessary to examine them in a Posner cueing procedure similar to that of Experiment 1. Our initial speculation is that the ears/beak side of the object (the left side in the orientation shown in Fig 12A), with its protruding distinctive features might be more salient and draw attention to that side of the object, and hence to targets on that side of space, as previously demonstrated by Leek & Johnston [27].

## Method

**Apparatus & design.** The apparatus and design were identical to that of Experiment 1 with the exception of the cue stimulus. Rather than using a teardrop shape, this experiment used a modified version of Fisher's [37] ambiguous figure (Fig 12A). The object appeared an equal number of times with the 'beak' to the left and to the right in a fully counterbalanced design of 112 trials. All stimuli and stimulus size details are available at osf.io/qxk8z.

**Participants.** Protocols approval, recruitment technique and recruitment criteria were identical to Experiment 1. Twenty-nine participants were tested but 26 remained following exclusions (see **Data exclusion and analysis**). Data was collected on the 20th December 2019.

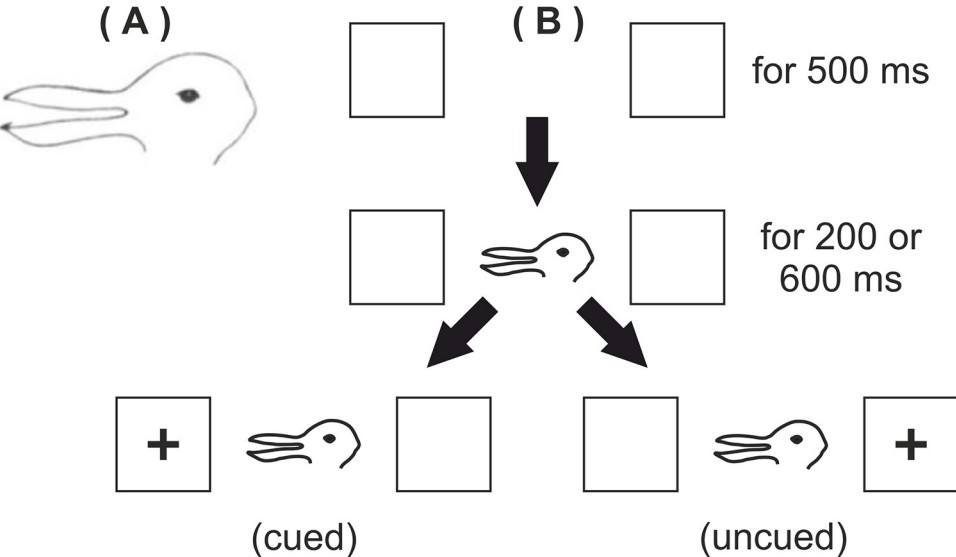

**Fig 12.** A) The ambiguous rabbit/seagull. B) Schematic representation of a trial in Experiment 4.

**Data exclusion and analysis.** Full exclusion details can be found at osf.io/qxk8z. Briefly: 1 participant was excluded due to error rate (>25% errors in either the target (i.e. >24) or distractor (i.e. >3) trials); and 2 participants were excluded due to few remaining trials following RT exclusion (<75% in any SOA × cue condition). The mean ± SD percentage of trials remaining in each SOA × cue for each participant 95.8 ± 5.5%.

## Results & discussion

Reaction times are shown in Fig 6. Bayesian repeated measures ANOVA on RTs with within subject factors of SOA (200/600 ms) and Cue (cued/uncued) support a model including both main terms ($BF_{10}$ = 5.730e+14 [*extreme evidence for H1*], $p$(H1|Data) = .761; SOA $BF_{incl.}$ = 1.001e+14; Cue $BF_{incl.}$ = 241.422). Reaction times were shorter for the 600 ms SOA than for the 200 ms SOA, and were shorter for the when the target was cued (seagull faced toward the target/rabbit faced away from the target) than when the target was uncued (seagull faced away from the target/rabbit faced toward the target). Frequentist modelling supports these findings. All models at osf.io/qxk8z.

The attention cueing experiment provides clear information concerning the properties of the rabbit/seagull stimulus. Targets are detected more rapidly when presented to the side of the ears/beak. This attention orienting is fast, within 200ms; stable as it is maintained up to 600ms; and automatic in that the rabbit/seagull stimulus does not predict the location of the up-coming target. As noted above, this asymmetric physical bias with distinctive features to one side of an object confirms that they can introduce directionality to an object representation, possibly evoked by attention capture by the features (e.g. Leek & Johnston [27]).

Therefore, as in the previous experiments, we have clear low-level properties that rapidly orient attention. The question in Experiment 5 is therefore whether priming a particular semantic interpretation can influence the third-party grouping in the visual search task. In that experiment one group of participants is presented with stimuli semantically associated with "rabbit" and the other with "seagull". For example, the rabbit group are told from the start they are involved in a rabbit experiment, they are shown a variety of drawings of rabbits in different postures, and in the search task they are to search for rabbits either facing towards or away from each other.

## Experiment 5 ('Seagull' identity priming)

### Method

**Apparatus.** The apparatus was identical to that of Experiment 2.

**Design.** The design was identical to that of Experiment 2 with two exceptions. Firstly, no social priming videos were shown. Secondly, the instruction screen for the main task block included the semantic prime for either rabbit or seagull by showing line drawings of the appropriate animal next to the instructions (see Fig 13A & 13B). For each participant the target shape in the search task was thereafter referred to as a 'rabbit' or 'seagull' as appropriate and the appropriate rabbit or seagull images where consistently observed before every block of search trials. All stimuli and stimulus size details are available at osf.io/qxk8z.

**Conditions.** Conditions were identical to Experiment 2.

**Participants.** Protocols were approved by the University of York's Psychology Departmental Ethics Committee and were in accord with the tenets of the Declaration of Helsinki. Participants were recruited through the University of York's Psychology Department participant recruitment system. Informed consent was obtained prior to participation. For the rabbit prime condition 30 participants were tested and 26 (age mean±SD = 19.6±1.4, 4 male) remained following exclusions (see Data exclusion and analysis). For the seagull prime

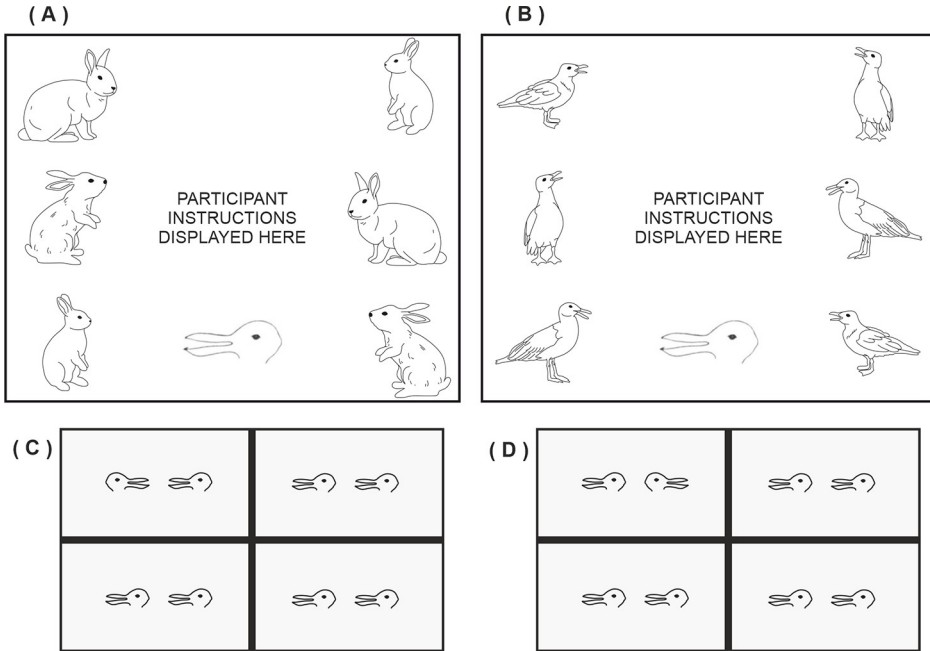

**Fig 13.** Examples of semantic priming on the instruction screen for the rabbit (A) and seagull (B) conditions in Experiment 5. Each of the six figures would appear randomly one by one. Schematic of the visual search task in a seagull inwards (C) and seagull outwards (D) display.

condition 30 participants were tested and 26 (age mean±SD = 21.1±7.8, 2 male) remained following exclusions (see Data exclusion and analysis).

**Data exclusion and analysis.** Full exclusion details can be found at osf.io/qxk8z. Briefly: 6 participants were excluded due to error rate (errors on >20% of trials); and 2 participants was excluded due to few remaining trials following RT and MT exclusion (<75% trials). The mean ± SD percentage of trials remaining in each condition for each participant was 95.6±8.4 in the rabbit prime condition and 97.4±4.3 in the seagull prime condition. Following all exclusion there were an equal number (n = 13) of participants in the A-B-A-B and B-A-B-A designs.

## Results & discussion

Bayesian repeated measures ANOVA on RTs (see Fig 10) with a within subjects factor of target orientation (seagull inwards/seagull outwards) and a between subjects factor of prime type (seagull face/rabbit face) support a model including only the target orientation term ($BF_{10}$ = 48.955 [*very strong evidence for H1*], $p$(H1|Data) = .448; $BF_{incl.}$ = 38.352). Reaction times were faster when detecting inwards pointing seagulls regardless of the prime.

In the target orientation task, Bayesian binomial tests (test value = 0.5) indicated that the majority of participants were able to identify the facing pair in both the seagull (22/26 participants, $BF_{+0}$ = 332.459 [*extreme evidence*]) and rabbit prime conditions (24/26 participants, $BF_{+0}$ = 15292.424, *extreme evidence*).

Frequentist modelling supports all findings. All models at osf.io/qxk8z.

Confirming our previous findings, this experiment demonstrates in the final object orientation question that our priming techniques are effective in creating an internal representation of object identity and facing direction, and we assume such representations are active during visual search. Nevertheless, low-level visual features that orient attention dominate and drive

the grouping effects when searching for targets amongst distractors. That is, independently of whether participants were primed to perceive rabbits or seagulls, the search data was equivalent: targets were detected significantly faster when the beaks/ears were oriented towards each other, which is exactly the result predicted from the prior attention cueing study and the results of Experiment 2 and 3.

## Experiment 6 ('Symmetrical shape' attention pre-test)

The data thus far clearly show that the low-level visual features of an object that evoke attention orienting determine the visual search performance. So far we have no evidence for higher-level social interaction experience influencing how the objects are represented and searched for. However, the results thus far do not unequivocally demonstrate that higher-level semantic representations of object identity and exposure to third-party social interactions are not represented.

For this issue, consider Fig 14, which represents Gestalt grouping processes. For most participants the image is initially grouped in terms of vertical columns, based on low-level grey-scale features. The higher-level subsequently computed horizontal grouping by shape (circle vs square) has little effect on how the display is grouped during the initial processing of the display. However, clearly the high-level shape information is internally represented, and indeed with further focussed processing, this structure can be extracted. The low-level earlier computed grey scale dominates initial attention capture and perceptual processing. Hence it is possible to make a similar claim in the experiments described thus far. It is possible that the third-

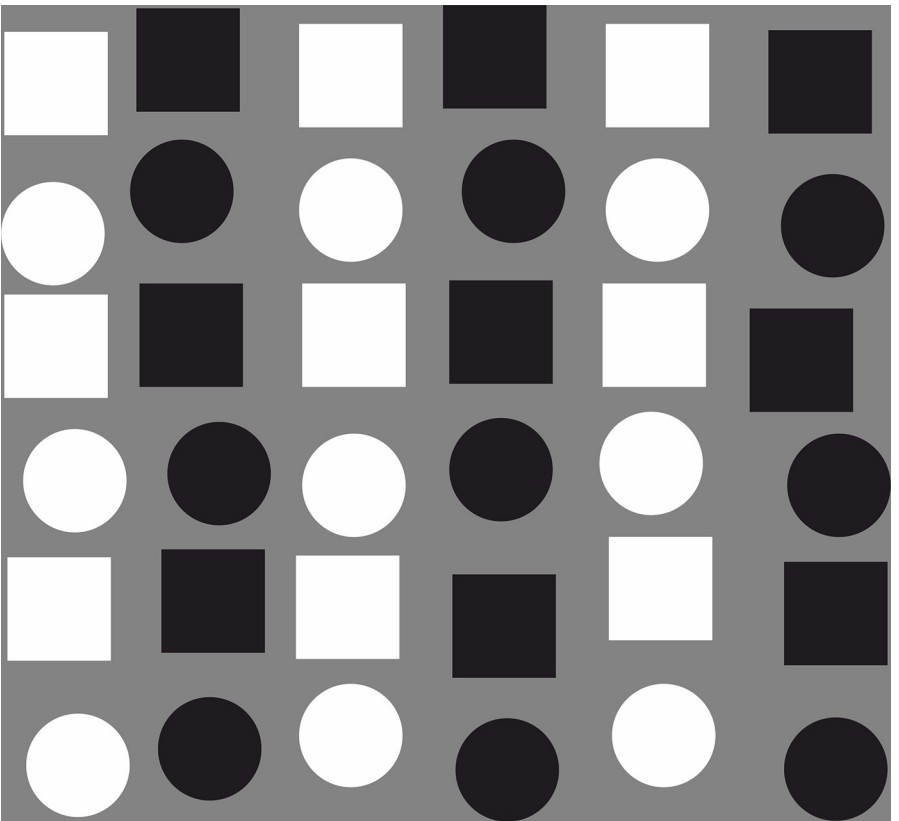

**Fig 14. Demonstration of Gestalt grouping.**

party social interaction relationships have been activated and are represented, but the low-level visual features, which are computed at earlier stages, dominate rapid search performance.

Indeed, this idea that encoding of social interactions might be a slow process that loses the race to more basic visual processes, has recently been supported by Isik et al. [38]. They report that the encoding of social interactions is a relatively late process taking around 300 to 500ms. This is in contrast to other complex visual processes such as face, object and scene processes that can be computed within 100 to 200ms. In our current experiments, the low-level features such as pointed ends that orient attention are even simpler than face, object and scene analysis, and hence can be encoded rapidly, further increasing the temporal contrast with the later third-party interaction processing.

Therefore, in our final experiments, we utilize stimuli that are symmetrical and do not have low-level visual features that orient attention to one side of space. This provides a further test of whether priming higher-level third party representations which bias which part of an object is forward facing, can influence grouping and performance in a visual search task.

However, before the visual search study, we have to investigate the attentional orienting properties of these new stimuli. That is, an initial study with these new symmetrical stimuli will ensure that there are no low-level visual properties that could bias attention to one side of space. For example, the stimuli differ in colour, and it is possible that one colour is more salient than the other. Hence we again employ the attention cueing task of Experiments 1 and 4, and now expect to see no attention orienting cueing effects. The lack of orienting effect will enable a further test of whether the higher-level third-party representations of the objects can be primed and influence search performance.

## Method

**Apparatus & design.** The apparatus and design were identical to that of Experiment 1 with the exception of the cue stimulus. Rather than using a teardrop shape, this experiment used a lemon shaped object that was two colours separated along its midline (see Fig 15). The object appeared an equal number of times with yellow on its left and on its right in a fully counterbalanced design of 112 trials. All stimuli and stimulus size details are available at osf.io/qxk8z.

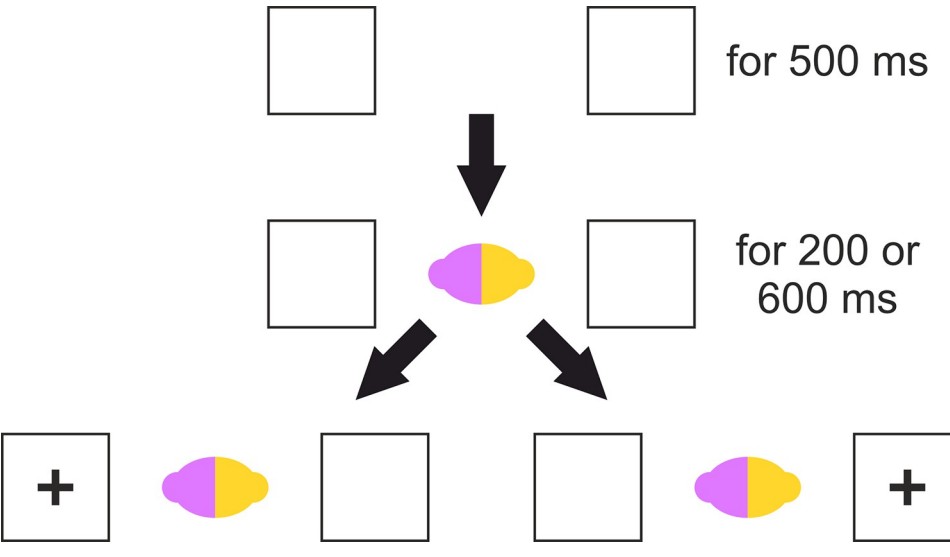

**Fig 15. Schematic representation of a trial in Experiment 6.**

**Participants.** Protocols approval, recruitment technique and recruitment criteria were identical to Experiment 1. Thirty-one participants were tested but 26 remained following exclusions (see Data exclusion and analysis). Data was collected on the 20th December 2019.

**Data exclusion and analysis.** Full exclusion details can be found at osf.io/qxk8z. Briefly: 3 participants were excluded due to error rate (>25% errors in either the target (i.e. >24) or distractor (i.e. >3) trials); and 2 participants were excluded due to few remaining trials following RT exclusion (<75% in any SOA × cue condition). The mean ± SD percentage of trials remaining in each SOA × congruency for each participant 96.3 ± 5.4%.

## Results & discussion

Reaction times are shown in Fig 6. Bayesian repeated measures ANOVA on RTs with within subject factors of SOA (200/600 ms) and Cue (yellow face/ purple face) support a model including only the SOA term ($BF_{10}$ = 2.343e+11 [*extreme evidence for H1*], $p$(H1|Data) = .790; SOA $BF_{incl.}$ = 1.634e+11). Reaction times were shorter for the 600 ms SOA than for the 200 ms SOA. Frequentist modelling supports these findings. All models at osf.io/qxk8z.

In contrast to Experiments 1 and 4, we find no evidence for attention cueing effects for either side of the stimulus. The lack of low-level visual features orienting attention ensures that this new stimulus provides an appropriate vehicle for testing whether higher-level representations of third-party interactions can be activated and influence visual search performance.

## Experiment 7 ('Symmetrical shape' social priming)

### Method

**Apparatus.** The apparatus was identical to that of Experiments 2 and 3.

**Design.** The design was identical to that of Experiment 3 with three exceptions. Firstly, the target changed from a teardrop shape to a symmetrical two-colour lemon shape from Experiment 6. Second, both coloured ends of the lemon shape could be primed as the "face" (as was the case in Experiment 2). Third, though the priming block remained, the video with targets changing colour was removed since the targets were already coloured.

Rather than introducing a white seal in the priming block, instead a purple/yellow seal was presented (Fig 16A). The between-subjects condition was whether the yellow or purple front was indicated as the face in this prime and in the two subsequent videos (Fig 16C & 16D). The motions and size of these targets were identical to the targets in Experiment 2 and 3. The colour change video from Experiment 2 was not used in the present experiment since the shapes would remain purple and yellow. After this initial semantic identity priming procedure, the social interaction videos containing 3 objects were presented before each search block as in Experiments 2 and 3. All stimuli and stimulus size details are available at osf.io/qxk8z.

**Conditions.** Conditions were identical to those of Experiment 2 (i.e. they differed from Experiment 3 in that either end (purple or yellow) of the target could be primed as the front.

**Participants.** Protocols were approved by the University of York's Psychology Departmental Ethics Committee and were in accord with the tenets of the Declaration of Helsinki. Participants were recruited through the University of York's Psychology Department participant recruitment system. Informed consent was obtained prior to participation. For the yellow face prime condition 37 participants were tested and 26 (age mean±SD = 19.6±1.4, 4 male) remained following exclusions (see Data exclusion and analysis). For the purple face prime condition 40 participants were tested and 26 (age mean±SD = 21.7±6.3, 4 male) remained following exclusions (see Data exclusion and analysis).

**Data exclusion and analysis.** Full exclusion details can be found at osf.io/qxk8z. Briefly: 21 participants were excluded due to error rate (errors on >20% of trials); and 4 participants

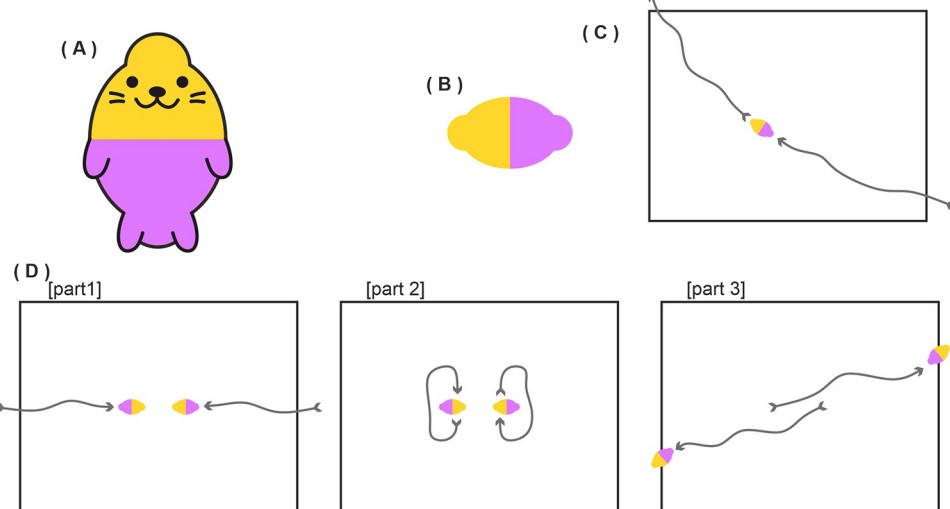

**Fig 16. Stimuli and primes in Experiment 7.** A) Seal prime. B) Seal target. C) Schematic representation of the first video prime in which a seal moves right to left. D) Schematic representation of the second video prime in which two seals approaching each other (part 1), greet each other (part 2) and then moving on (part 3). All stimuli available at osf. io/qxk8z.

were excluded due to few remaining trials following RT and MT exclusion (<75% trials). The mean ± SD percentage of trials remaining in each condition for each participant was 95.8±8.9 in the yellow face prime condition and 97.7±6.6 in the purple face prime condition. Following all exclusion there were an equal number (n = 13) of participants in the A-B-A-B and B-A-B-A designs of each prime condition.

## Results & discussion

Bayesian repeated measures ANOVA on RTs (see Fig 10 with a within-subjects factor of target orientation (yellow inwards/yellow outwards) and a between-subjects factor of prime type (yellow front/purple front) does not support any model over the null ($BF_{10} < = .757$). Reaction times did not differ between conditions.

In the task where participants judged which target pairs were facing each other, Bayesian binomial tests (test value = 0.5) indicated that the majority of participants were able to identify the facing pair in both the yellow (21/26 participants, $BF_{+0} = 75.513$ [*very strong evidence*]) and purple prime conditions (25/26 participants, $BF_{+0} = 191193.305$, [*extreme evidence*]).

Frequentist modelling supports all findings. All models at osf.io/qxk8z.

The results confirm our previous observations. First, priming techniques were effective at communicating target front even with symmetrical targets. Second, even after extensive repeated experience of object identities and observing object social interactions, the representations do not influence performance on a visual search task. For example, if a participant had been exposed to a series of events showing that the yellow end of an object has agency and is equivalent to the "face" during a series of video displays, detection of towards facing yellow stimuli is not facilitated during search. This result confirms the findings of Experiments 2, 3, and 5.

## General discussion

Previous research has shown that in complex and cluttered environments participants are able to detect interacting people faster than non-interacting people (e.g. [1,2,12]). This process

would appear to be a valuable way to parse complex social scenes into interacting individuals where important social processes might be taking place, which would be worthy of further analysis to interpret the scene and predict potential future behaviours. For example, when two people are perceived to be interacting, the emotion of one influences the perceived emotion of the other (e.g. [39]) and subsequent short- and longer-term memory of the interaction is influenced by the initial computation of the social interaction [1]. The issue we have examined here concerns what the specific mechanisms might be that mediate this initial processing that enables the structuring of social scenes.

On the one hand, there may be rapid encoding of the third-party interaction between social beings. Such encoding of high-level interpersonal processes might predict that the visual search effects would only be detected when observing social animals such as humans. Indeed, Vestner et al. [1] explicitly argued that effects were caused by higher-level representations of social interactions, rather than lower level perceptual processes. However, subsequent work (e.g. [12]) has challenged this conclusion, at least in visual search tasks, by demonstrating that in fact the same social priority effects in visual search can be detected when participants search for towards vs away facing arrows. As these arrow stimuli do not possess the social properties of biological systems, this would suggest that the effect is driven by a non-social low-level attention orienting process (e.g. [8]). The interpretation in terms of general attention mechanisms is that inwards faces or arrows orient the beam of attention to one central location between the two critical target objects facilitating search. In contrast, away facing people or arrows evoke attention shifts in opposite and hence competing directions. Such splitting of attention would impair the judgment of the relationship between the two objects.

However, although the effects demonstrated with arrows are equivalent to those of faces, suggesting that higher-level social representations of interacting individuals are not necessary to produce the effects, they do not unequivocally demonstrate that higher-level social processes are not involved due to potential ceiling effects preventing the detection of additive effects. And indeed, a range of studies have in fact argued for the role of mentalizing processes during social shifts of attention, such as learning of trust (e.g. [40]) and action intention (e.g. [41], and [42] for review). Therefore, in this series of studies we have investigated this issue further. A series of experiments using a range of converging methods has examined whether creating higher-level representations of objects as interacting individuals could influence the initial structural encoding of visual scenes, independently of low-level visual properties. A wide range of previous research has demonstrated the potency of the video priming techniques we have used, where effects can be observed in pre-language 6-month olds for example [24]. And we confirmed that such techniques appear to influence the representation of social agency in the direction of the object's attention. The evidence is clear within this research programme that such higher-level representations of social interactions created by our social priming techniques are not playing a significant role.

Clearly, as this is a null finding, we have to be cautious with our interpretations. Nevertheless, further analysis combining Experiments 2, 3 and 5 provides substantial power to support our conclusions. Within this analysis, there is substantial evidence that our visual search effects are dominated by low-level visual features that automatically trigger attention orienting ($BF_{10}$ = 2.435e+9 [*extreme evidence for H1*], $p$(H1|Data) = .687; $BF_{incl.}$ = 1.784e+9), and any inclusion of social priming considerably worsens model fitting by a factor of 15 or greater (see model at osf.io/qxk8z).

As discussed, we are not discounting higher-level social processes in all situations, and indeed in the Vestner et al. [1] studies, it is possible that such effects are taking place at the later processing stages of working memory maintenance and retrieval from longer-term memory. Rather, we argue it is likely that they play a limited role in the earliest stages of processing,

where there is rapid structuring of the visual environment to create internal representations for further processing. Thus, we suggest that the higher-level mentalizing processes may be at play at later stages as suggested by evidence from neuroscience [38] and behavioural studies showing that the effects of mentalizing on gaze following are slower non-automatic processes (e.g. [43], though also note recent work potentially indicating mentalising as a fast automatic process e.g. [44,45]). As an example of two potential processes, one rapid and another slower, consider the attention cueing study of [46]. In that study the rapid attentional cueing effects evoked by gazing faces, as measured by reaction time to detect peripheral targets, were unaffected by the emotion of the face (smile vs disgust). However, in the same task a slower social learning process was simultaneously at play, where subsequent decisions concerning object liking were influenced by the interaction between gaze and emotion. In a similar vane, tasks requiring deeper encoding where participants actively switch perception from their own first-person to another's third-person perspective before judging a visual scene, also provide evidence for slower mentalizing processes [47].

In conclusion, rapid parsing of social scenes into interacting vs non-interacting individuals is an important process that provides initial representations for further more sophisticated processing. However, our data suggest that this initial process structuring the social world is not based on complex and sophisticated representations of socially interacting individuals. Rather, the highly efficient attention systems utilizing basic perceptual features that have evolved for the interaction between vision and action to enable selective goal-directed action can also serve these social computations. The employment of basic attention processes would appear to be the most parsimonious and efficient way of rapidly structuring visual inputs to reflect social interactions.

## Acknowledgments

We would like to thank Bryony G. McKean and Edward Hindmarsh for assistance with data collection.

## Author Contributions

**Conceptualization:** Jonathan C. Flavell, Harriet Over, Tim Vestner, Steven P. Tipper.

**Data curation:** Jonathan C. Flavell.

**Formal analysis:** Jonathan C. Flavell.

**Funding acquisition:** Harriet Over, Steven P. Tipper.

**Investigation:** Jonathan C. Flavell, Harriet Over, Steven P. Tipper.

**Methodology:** Jonathan C. Flavell, Harriet Over, Tim Vestner, Steven P. Tipper.

**Project administration:** Jonathan C. Flavell, Steven P. Tipper.

**Resources:** Jonathan C. Flavell.

**Software:** Jonathan C. Flavell.

**Supervision:** Harriet Over, Steven P. Tipper.

**Visualization:** Jonathan C. Flavell, Steven P. Tipper.

**Writing – original draft:** Jonathan C. Flavell, Harriet Over, Tim Vestner, Richard Cook, Steven P. Tipper.

**Writing – review & editing:** Jonathan C. Flavell, Harriet Over, Tim Vestner, Richard Cook, Steven P. Tipper.

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
