## [Decision Letter · Decision Letter 0]

26 Apr 2021

PONE-D-21-06244

Rapid detection of social interactions is the result of domain general attentional processes

PLOS ONE

Dear Dr. Flavell,

Thank you for submitting your manuscript to PLOS ONE. After careful consideration, we feel that it has merit but does not fully meet PLOS ONE’s publication criteria as it currently stands. Therefore, we invite you to submit a revised version of the manuscript that addresses the points raised during the review process.

Three independent experts commented on your manuscript. One of them disclosed as Hauke Meyerhoff. As you can see from the reviews, all referees found the general topic addressed in your manuscript interesting and they have many nice things to say about the study. At the same time, they have a whole number of remarkably constructive and excellently detailed suggestions how to further improve the paper. The comments speak for themselves, but it is obvious that one reoccurring theme is the need for more specificity regarding the theoretical concepts., and some minor methodological issues. While this will call for some extra efforts, I consider it worthwhile. Therefore, we invite you to submit a revision of the manuscript that addresses the remaining points together with a cover letter that contains point-by-point replies.

We look forward to receiving your revised manuscript.

Kind regards,

Michael B. Steinborn, PhD

Academic Editor

PLOS ONE

Journal Requirements:

Reviewers' comments:

Reviewer's Responses to Questions

**Comments to the Author**

1. Is the manuscript technically sound, and do the data support the conclusions?

Reviewer #1: Yes

Reviewer #2: Yes

Reviewer #3: Partly

2. Has the statistical analysis been performed appropriately and rigorously? 

Reviewer #1: Yes

Reviewer #2: Yes

Reviewer #3: Yes

3. Have the authors made all data underlying the findings in their manuscript fully available?

Reviewer #1: Yes

Reviewer #2: Yes

Reviewer #3: Yes

4. Is the manuscript presented in an intelligible fashion and written in standard English?

Reviewer #1: Yes

Reviewer #2: Yes

Reviewer #3: Yes

5. Review Comments to the Author

Reviewer #1: This is interesting research. The authors conducted seven experiments that systematically assess the influence of basic perceptual features vs. that of higher-level social representation on visual search. The authors make data, results and stimuli publicly available. They report Bayesian analyses in the main text but also attach frequentist statistics in the online supplemental materials.

I really enjoyed reading this paper and many of my remarks are intended to improve readability. However, I also have some suggestions for extensions of this manuscript

General remarks:

- Experiment 1/4/6 were conducted online while experiments 2, 3, 5 and 7 were conducted in the lab. Most likely, the participants of experiments 1/4/6 vs. 2/3/5/7 stem from different cohorts. I don’t think that this is a big issue since the authors don’t directly compare results from these experiments, but I still think that this should be mentioned briefly in the discussion

- In the document containing the models at OSF, it is not readily apparent what the factor levels refer to. E.g. in experiment 2, there is the factor “TA” with the levels “T” and “A”. Another example: In experiment 6, there is a factor labeled “congruency”. What does this name refer to when there were no directional cues in this experiment? Could the authors please rename each factor (level) in each experiment so that their meaning is obvious? Please also go through all headings and captions and check whether everything is self-explanatory. For example, it is not clear to me what “MTs” refers to (e.g. in “ANOVA on MTs”). Please clarify this, e.g. in the caption.

- Also, in the model document in the osf respository, it is now the case that the model solution containing the interaction term is printed in bold (e.g., in the first table, the solution with two main effects plus the interaction term (“SOA + Congruency + SOA * Congruency”) is printed in bold although this does not provide the best solution). This is a bit misleading as I would expect the best solution to stick out. My suggestion is to either highlight the best model or none

Introduction:

I found the introduction well understandable and straight-forward. The hypotheses are clear and plausible. I have two small suggestions:

- In the introduction, the authors make a point about parallel processing streams for social and directional cues. Perhaps this could be strengthened by citing Ristic, Friesen & Kingstone (2002), Psychonomic Bulletin & Review (e.g. on page 5, lines 81 ff.)

- I found the experiment overview figure in the OSF database very helpful to keep track/get an overview of the setup of the seven experiments. Perhaps this figure could also be included in the manuscript

Generally, I found the descriptions of the experiments clear and easy to follow. The results are convincing and I agree with the author’s interpretations. I have some remarks regarding some of the experiments:

Experiment 1:

- Maybe the authors could add on page 11 para 1 whether the participants were explicitly instructed how to press the respond keys (e.g. both index fingers or two fingers of the same hand)

Experiment 2:

- Could the authors please report the visual angle of stimulus presentation? SR research provides a nice tool to calculate it: https://www.sr-research.com/visual-angle-calculator/

- Is there a specific reason why, in Figure 7, the upper two left-hand boxes say ‘with round as “face”’ while the lower two say ‘(round as “face”)’? If not, I suggest to chose either one and keep it consistent

- I presume that this was just a technical error in the uploading process, however, the axis labels of Figure 9 are very hard to read. Also, I suggest no to divide e.g. 1200 ms with a comma (1,200 ms)

- Could the authors please add short descriptives to the main text that indicate the direction of an effect (e.g. lines 379-380: were inwards or outwards facing targets responded to faster?)

Experiment 3

- I think, Figures 10 and 11 were mixed up in the submission. In my PDF, Figure 10 depicts the seagull/rabbit Figure and Figure 11 shows the seal

Experiment 4

- No remarks

Experiment 5

- Since the participants were Psychology students, I wonder whether they (or some of them) might have been aware of the rabbit/duck figure. Did the authors assess this (e.g. by asking participants about it, or how they perceived the figure) and exclude participants who were familiar with it? If yes, this should be reported as well. If not, I don’t think it is much of a problem because binomial tests confirm that the cues were perceived as intended

Experiment 6

- No remarks

Experiment 7

- The number of participants excluded due to error rates is much higher in this experiment compared to experiment 1-6 (11 and 14 compared to 3-6 participants). Do the authors have any idea about why this is the case?

General Discussion

As the rest of the manuscript, the general discussion is neat and well-readable. I have two points to add:

- I miss a little bit of a summary of the main findings of the experiments and how they are interrelated / what big picture they form. This would be particularly helpful for readers who don’t have time to read the whole paper in detail

- On page 38, the authors discuss the interpretation of null results and introduce new results in this section. I feel that this paragraph is worth extension and more results could be included. Perhaps this could be done (in part) in a separate section prior to the general discussion

Minor points:

Line 769: in a similar vein

Reviewer #2: Review for Flavell et al. (submitted to PlosOne). Rapid detection of social interactions is the result of domain general attentional processes

Summary

The authors present a big set of experiment investigating whether low-level perceptual features of high-level social features guide attentional processing in visual search tasks. They test shapes with orientation information, schematic animals (ambiguous drawings), as well as neutral shapes with differently colored ends. The pattern of results is consistent with the interpretation that low-level properties guide attention.

Evaluation

This is a good set of experiments. I only have a couple of points and references which I think can improve the manuscript. I want to explicitly emphasize that I appreciated the way how the authors managed to keep the manuscript short despite reporting so many experiments.

Major Comments

1. One shortcoming in the experiments is that the authors do not compare the different stimuli within the same experiment. Such a design would allow to test for interactions. Instead, they report the presence of an effect in one experiment and the absence of an effect in another experiment (which does not allow the conclusion that both effects differ from each other). I think this should be mentioned and discussed. Given the rather strong results, however, I do not think that this needs to be run as an additional experiment.

2. There is a related field of research which came up with matching findings which the authors might find interesting to broaden their discussion (which is rather narrow). Visual search and attentional guidance research on perceptual animacy has shown that rather low-level properties rather than the social properties guide the detection of such interactions (Meyerhoff, Schwan, & Huff, 2014, JEP:HPP; Meyerhoff, Schwan, & Huff, 2014; PB&R). In the same way, the authors might find the literature on the wolfpack effect interesting (Gao, MacCarthy, & Scholl, 2010, Psych. Sci.)

3. The authors report the BF10 for the final models (i.e. for the factors which explain the results). As a part of the story is to say that the other factors (e.g. the priming manipulations) are irrelevant for explaining the results. It would be nice to see the BF01 or BF10 for these factors as well (maybe more general the BFs for factors rather than models?).

Minor Stuff

- There seem to be some residuals from previous submissions (e.g., there always will be color in an online journal) which should be removed from the ms.

- Figure 11 is missing

- Stimuli sizes are probably important here, so they should be in the Methods.

Signed,

Hauke Meyerhoff

Reviewer #3: This paper seemed great in several ways: the question motivating the study is interesting and important, the results are clear and robust, and it's a well-written and well-organized manuscript. I also find this series of experiments to be a nice addition to the authors’ past work, and admire the depth of the research program as a whole. Along with my enthusiasm, I have a few concerns about the interpretation and discussion of the results, as well as some minor comments.

Major comments

1. A crucial assumption underlying the conclusions is that the videos primed higher-level social interpretations; this was assessed via a final forced-choice question asking participants to select which stimuli were facing each other. Based on responses largely congruent with the primes, the authors conclude they have successfully manipulated “the representation of social agency in the direction of the object’s attention” (p.38). But the DV (i.e., participants’ ability to select the front as instructed) seems different from the conclusions (i.e., participants’ representations of social properties/joint attention). First of all, this forced-choice question is extremely susceptible to demand characteristics. For example, in Expts. 3/7 it would have been very surprising if participants had indicated the pointy/purple end of the seal as its front after seeing the speech bubble coming out of its round/yellow end. This is even more apparent in Expt. 5, where the instructions prompted participants to “pick which pair of seagulls are facing each other”; the fact the intended interpretation of the stimuli is re-iterated in the question makes me wonder whether participants were simply selecting what they had been taught was the correct option – regardless of their actual percepts. So I am on board when the authors label these primes as “semantic labels” (p. 26, 34), but not when they describe them as directly manipulating “the representation of social agency in the direction of the object’s attention” (p.38) or “high-level representations of social interactions” (p.38), or when equating these manipulations to the perception of a face or social interactions – which seems fundamentally different in nature.

2. Even assuming the primes worked as intended, I wonder if the relevant interpretations were active during the search task itself. Perceptual interpretations of ambiguous figures have been shown to rely on fixation patterns and covert attention (e.g. Peterson & Gibson, 1991, JEP:HPP; Toppino, 2003, P&P) – but both fixations and covert attention are key for successful visual search, and so top-down interpretations of ambiguous stimuli are directly in competition with search performance (which is especially high here). This is not a problem in itself, but it highlights another problem with generalizing from ambiguous stimuli to faces and people, since face perception doesn't require top-down control (or not even awareness, e.g. Stein et al., 2012, Cognition).

3. Another assumption here is that default percepts of fronts reflect lower-level visual features; but I don't think this can be taken for granted. For example, the fact that seagulls facing each other are prioritized in Expt. 5 could be an effect of the interpretation of seagulls as seagulls (in both conditions, if the prime is inactive as per points #1 and #2) and it is in fact their perceived facing direction that is driving attention. (Relatedly, the claim on p.7 that the stimuli “contain no intrinsic visual features that could be construed as a face” and have “no salient intrinsic face-like features” doesn’t seem right to me, since the seagulls/rabbits have a beak, ears, and eye.) This also applies to Expt. 2, as some authors have suggested that arrows can in fact be socially meaningful (e.g. Kingstone et al., 2003) or suggest an agentic presence, especially when multiple stimuli point towards the same region of space (e.g. Gao et al., 2010, Psych Sci; Takahashi et al., 2013, Front Hum Neurosci; Colombatto et al., 2019, Perception). Since these results as described have potential implications for that literature, it would be great to see this discussed.

4. One of the reasons social binding was initially thought to be a social effect was that it vanished with inverted faces that are equivalent to upright faces in lower-level properties, but differ in the higher-level social properties (although related to my point #1, participants would still be able to indicate whether people are facing each other, even when they are inverted!). Those findings contradict the conclusion that lower-level properties only drive social binding – and since this work directly follows up on those experiments, it would be interesting to see the authors discuss this.

Minor comments

1. The main text for both Expts. 6 and 7 describes a ‘yellow’ vs. ‘purple’ manipulation (also depicted in Figs. 5, 14-15). But then Fig. 9 depicts ‘orange’ vs. ‘purple’ conditions, and the datafiles report ‘orange’ and ‘purple’ for Expt. 6, and ‘rabbit’ and ‘orange’ for Expt. 7 (in both the RT and MT files); could these be made consistent?

2. How were subjects assigned to the prime conditions – were they alternating? (This would be helpful to clarify as potentially related to block order assignment.)

3. How were the prime videos generated for the ‘round-face’ conditions? Were the stimuli simply mirrored/rotated? I ask because this seems to have produced some artifacts, e.g. the ‘round-faces’ overlap at the end of the videos and during the ball tossing game, which would be weird for agents, or the ‘round-face’ stimuli either toss the ball at a distance (rightmost), or overlap with the ball (leftmost).

4. The target orientation question for Exp. 2 is phrased in the instructions as “You need to pick which pair of seals are facing each other”, but I don’t think ‘seals’ was ever used elsewhere in Exp. 2. How could participants answer this question?

5. This statement seems inaccurate/controversial: “higher-level mentalising processes are at play at later stages as suggested by […] behavioural studies showing that the effects of mentalising on gaze following are slower non-automatic”: the matter is still debated, but recent evidence showing the contrary should be cited, e.g. that mentalising such as perspective taking can occur rapidly and automatically (e.g. Ward et al., 2019, Curr Bio), and that percepts of mental states can influence lower-level processes such as gaze cueing, even quickly and automatically (e.g. Colombatto et al., 2020, PNAS).

6. For additional evidence that the pointy end of the teardrop is seen as its front, the authors may find helpful Chen & Scholl, 2018, PB&R – who used teardrop stimuli to elicit the same effects (on aesthetic preferences) as other types of fronts. (I wouldn't normally mention too many papers from our group in a review, but each one here seems warranted as potentially helpful or directly relevant to the discussion.)

6. PLOS authors have the option to publish the peer review history of their article (what does this mean?). If published, this will include your full peer review and any attached files.

Reviewer #1: **Yes: **Christina Breil

Reviewer #2: No

Reviewer #3: No

---

## [Author Response · Author response to Decision Letter 0]

16 Aug 2021

RESPONSE TO REVIEWERS

NOTE - copied from the uploaded document 'Response to Reviewers_v2.docx. Formatting may vary. 

Dear Dr Steinborn,

Thank you for your letter concerning our paper “Rapid detection of social interactions is the result of domain general attentional processes”. We are grateful for the opportunity to resubmit our revised manuscript. As always, we appreciate the time and effort you and the reviewers have committed to our work, and we were very pleased by the positive and helpful feedback. In the following we describe our response to the reviewer feedback. Reviewer comments are shown in black and our responses are shown in blue.

Yours sincerely,

Jonathan Flavell, Harriet Over, Tim Vestner, Richard Cook, Steven Tipper

-o0o-

REVIEWER #1

This is interesting research. The authors conducted seven experiments that systematically assess the influence of basic perceptual features vs. that of higher-level social representation on visual search. The authors make data, results and stimuli publicly available. They report Bayesian analyses in the main text but also attach frequentist statistics in the online supplemental materials.

I really enjoyed reading this paper and many of my remarks are intended to improve readability. However, I also have some suggestions for extensions of this manuscript

Thank you for these kind comments.

General remarks:

Experiment 1/4/6 were conducted online while experiments 2, 3, 5 and 7 were conducted in the lab. Most likely, the participants of experiments 1/4/6 vs. 2/3/5/7 stem from different cohorts. I don’t think that this is a big issue since the authors don’t directly compare results from these experiments, but I still think that this should be mentioned briefly in the discussion

We have now raised this in ‘Participants’ of Experiment 2.

In the document containing the models at OSF, it is not readily apparent what the factor levels refer to. E.g. in experiment 2, there is the factor “TA” with the levels “T” and “A”. Another example: In experiment 6, there is a factor labeled “congruency”. What does this name refer to when there were no directional cues in this experiment? Could the authors please rename each factor (level) in each experiment so that their meaning is obvious? Please also go through all headings and captions and check whether everything is self-explanatory. For example, it is not clear to me what “MTs” refers to (e.g. in “ANOVA on MTs”). Please clarify this, e.g. in the caption.

Also, in the model document in the osf respository, it is now the case that the model solution containing the interaction term is printed in bold (e.g., in the first table, the solution with two main effects plus the interaction term (“SOA + Congruency + SOA * Congruency”) is printed in bold although this does not provide the best solution). This is a bit misleading as I would expect the best solution to stick out. My suggestion is to either highlight the best model or none

Thank you for taking the time to check the ‘Models’ document. We apologise for the unclear labelling and the misleading bold type. A new ‘Models’ document has been uploaded to the OSF with un-bolded models (this was a formatting error in the first upload) and labels changed to match the terminology used in the manuscript. See below for label changes (from → to): 

• Congruency → Cue

o Con → Cued

o Incon → Uncued; 

• TA → Orientation

o T → Inwards

o A → Outwards

• RT → Reaction Time

• MT → Movement Time

Introduction:

I found the introduction well understandable and straight-forward. The hypotheses are clear and plausible. I have two small suggestions:

- In the introduction, the authors make a point about parallel processing streams for social and directional cues. Perhaps this could be strengthened by citing Ristic, Friesen & Kingstone (2002), Psychonomic Bulletin & Review (e.g. on page 5, lines 81 ff.)

Thank you for the helpful reference. We’ve added it to the introduction as suggested.

- I found the experiment overview figure in the OSF database very helpful to keep track/get an overview of the setup of the seven experiments. Perhaps this figure could also be included in the manuscript

We have also added the experiment overview figure to the introduction. 

Generally, I found the descriptions of the experiments clear and easy to follow. The results are convincing and I agree with the author’s interpretations. I have some remarks regarding some of the experiments:

Experiment 1:

- Maybe the authors could add on page 11 para 1 whether the participants were explicitly instructed how to press the respond keys (e.g. both index fingers or two fingers of the same hand)

Participants were not instructed on which fingers to use for the task (we have now indicated this in the manuscript) but in piloting participants always used their left index finger for ‘F’ and their right index for ‘J’ on the lab QWERTY keyboard. 

Experiment 2:

- Could the authors please report the visual angle of stimulus presentation? SR research provides a nice tool to calculate it: https://www.sr-research.com/visual-angle-calculator/

Participants were not head rested and head position was not so horizontal and vertical visual angles are not available. The reported viewing distance of ~50cm from the screen is intended only as a guide to set-up. 

- Is there a specific reason why, in Figure 7, the upper two left-hand boxes say ‘with round as “face”’ while the lower two say ‘(round as “face”)’? If not, I suggest to chose either one and keep it consistent

Thank you for pointing this out. All items were supposed to be identical. This has been corrected. Note that this is now Figure 8.

- I presume that this was just a technical error in the uploading process, however, the axis labels of Figure 9 are very hard to read. Also, I suggest no to divide e.g. 1200 ms with a comma (1,200 ms).

Axis labels in the uploaded .tiff file appear correct so we agree that it must be an upload error. We have removed the comma from y axis labels. A png version of the image is below in case a re-submitted figure has the same issue. Note that this is now Figure 10.

- Could the authors please add short descriptives to the main text that indicate the direction of an effect (e.g. lines 379-380: were inwards or outwards facing targets responded to faster?)

Thank you for suggesting this. We have added a single sentence to the end of each model description of the main text to describe the outcome in terms of difference in RT by condition. 

Experiment 3

- I think, Figures 10 and 11 were mixed up in the submission. In my PDF, Figure 10 depicts the seagull/rabbit Figure and Figure 11 shows the seal

Thank you for spotting this. It was an error in submission. We will double check figure file labels on re-submission. 

Experiment 4

- No remarks

Experiment 5

- Since the participants were Psychology students, I wonder whether they (or some of them) might have been aware of the rabbit/duck figure. Did the authors assess this (e.g. by asking participants about it, or how they perceived the figure) and exclude participants who were familiar with it? If yes, this should be reported as well. If not, I don’t think it is much of a problem because binomial tests confirm that the cues were perceived as intended

One of the authors demonstrates the rabbit/duck figure in a lecture series every year so this had been considered. The experiment was conducted early in the term before this lecture took place. Participants were not asked about familiarity with the figure. 

Experiment 6

- No remarks

Experiment 7

- The number of participants excluded due to error rates is much higher in this experiment compared to experiment 1-6 (11 and 14 compared to 3-6 participants). Do the authors have any idea about why this is the case?

The number of exclusions due to error rate was indeed higher in Experiment 7 than it was using the same design with different stimuli (Experiment 2, 3 and 5). The majority of all participants excluded for error rate have exceptionally poor performance generally due to, anecdotally, not understanding the task or lack of effort. Experiment 7 did not differ in that respect but 8 of the excluded participants had error rates at ~30% or less so under less stringent conditions their data may have been included. The remaining participants typically had error rates >75%. We believe that the relatively higher error in Experiment 7 is likely due to stringent performance requirements to ensure data quality coupled with a few less diligent participants. 

General Discussion

As the rest of the manuscript, the general discussion is neat and well-readable. I have two points to add:

- I miss a little bit of a summary of the main findings of the experiments and how they are interrelated / what big picture they form. This would be particularly helpful for readers who don’t have time to read the whole paper in detail.

Thank you for this excellent suggestion. We appreciate that the article is quite complex with a range of different approaches used in 7 experiments. In light of this we have now included a preview of our findings and a new figure 4 at the end of the Introduction.

- On page 38, the authors discuss the interpretation of null results and introduce new results in this section. I feel that this paragraph is worth extension and more results could be included. Perhaps this could be done (in part) in a separate section prior to the general discussion

Features were found to dominate attention in all cases and this new combined analysis is intended only to lend credence to those findings. As a result, we have sought to keep the discussion relatively brief. 

Minor points:

Line 769: in a similar vein

We are not sure what the reviewer is referring to but we have briefly expanded on the paper by Furlanetto et al. 

-o0o-

REVIEWER #2

Review for Flavell et al. (submitted to PlosOne). Rapid detection of social interactions is the result of domain general attentional processes

Summary

The authors present a big set of experiment investigating whether low-level perceptual features of high-level social features guide attentional processing in visual search tasks. They test shapes with orientation information, schematic animals (ambiguous drawings), as well as neutral shapes with differently colored ends. The pattern of results is consistent with the interpretation that low-level properties guide attention.

Evaluation

This is a good set of experiments. I only have a couple of points and references which I think can improve the manuscript. I want to explicitly emphasize that I appreciated the way how the authors managed to keep the manuscript short despite reporting so many experiments.

Thank you for this kind and generous evaluation of our work.

Major Comments

1. One shortcoming in the experiments is that the authors do not compare the different stimuli within the same experiment. Such a design would allow to test for interactions. Instead, they report the presence of an effect in one experiment and the absence of an effect in another experiment (which does not allow the conclusion that both effects differ from each other). I think this should be mentioned and discussed. Given the rather strong results, however, I do not think that this needs to be run as an additional experiment.

Re. Posner cueing experiments. We believe that including the teardrop, seagull, symmetrical shape stimuli in a single Posner cueing task is likely to reveal similar results to those found in the three separate tasks – principally that shape direction strongly orients attention to one side or another. 

Re. Visual search experiments. We assume the the reviewer is considering a 2x2 grid visual search task featuring symmetrical shape pairs and non-symmetrical shape pairs with either the shape type being primed as a social agent beforehand. Again, we believe that similar results to the current experiment would likely be found – faster responses when detecting inwards pointing pairs with no prime effect. 

However, we agree that it could be interesting to pursue a line work comparing the influence of asocial directional cues (teardrop) with social directional cue (seal and seagull) and social non-direction cues (symmetrical shape) using primes for one or more types. 

2. There is a related field of research which came up with matching findings which the authors might find interesting to broaden their discussion (which is rather narrow). Visual search and attentional guidance research on perceptual animacy has shown that rather low-level properties rather than the social properties guide the detection of such interactions (Meyerhoff, Schwan, & Huff, 2014, JEP:HPP; Meyerhoff, Schwan, & Huff, 2014; PB&R). In the same way, the authors might find the literature on the wolfpack effect interesting (Gao, MacCarthy, & Scholl, 2010, Psych. Sci.)

Thank you for these interesting articles. We have now mentioned this work in the Introduction, in set the scene for our new experiments.

3. The authors report the BF10 for the final models (i.e. for the factors which explain the results). As a part of the story is to say that the other factors (e.g. the priming manipulations) are irrelevant for explaining the results. It would be nice to see the BF01 or BF10 for these factors as well (maybe more general the BFs for factors rather than models?).

From each Bayesian ANOVAs we report only the ‘best’ (the model with the largest BF10) model in the manuscript. This does not mean that all other models (with different factors or different combinations of factors) are null. 

For example, in the table below (copied from the supplementary file ‘Models.docx’ at www.osf.io/qxk8z) all the models fit the data better than the null model though the extent of this varies between models. A model with the just the Cue factor is rather poor whereas the model including SOA + Cue factors and the model including the SOA + Cue + SOA*Cue factors predict the data extremely well. In the manuscript we report the SOA + Cue model because it fits the data ~2.25 times better than the SOA + Cue + SOA*Cue model. 

As described in the Results sections, all models are available open access at www.osf.io/qxk8z. To keep the manuscript succinct and convey the relevant points clearly, we would prefer to keep the ‘rejected’ models and that supplementary material. 

Note that BF01 can be found as 1/BF10. So a BF10=3.005e+7 (an extremely large number) is equal to BF01=4.519e-4 (an extremely small number).

Model Comparison 

Models P(M) P(M|data) BF M BF 10 error % 

Null model (incl. subject) 0.200 2.005e -8 8.021e -8 1.000 

SOA + Cue 0.200 0.603 6.065 3.005e +7 2.489 

SOA + Cue + SOA  ✻  Cue 0.200 0.268 1.466 1.338e +7 2.449 

SOA 0.200 0.129 0.594 6.444e +6 4.574 

Cue 0.200 2.840e -8 1.136e -7 1.417 1.153 

Note. All models include subject 

Analysis of Effects 

Effects P(incl) P(incl|data) BF incl 

SOA 0.600 1.000 1.376e +7 

Cue 0.600 0.871 4.493 

SOA  ✻  Cue 0.200 0.268 1.466 

Minor Stuff

- There seem to be some residuals from previous submissions (e.g., there always will be color in an online journal) which should be removed from the ms.

Thank you for spotting this error, we have corrected it. 

- Figure 11 is missing

In our PLOSONE generated pdf version of the manuscript all of the figures are presented at the end of the manuscript but the order is slightly off… figure 8, figure 9, figure 11, figure 10, figure 12. 

- Stimuli sizes are probably important here, so they should be in the Methods.

We have added details of stimulus sizes and relative positions to a separate document on the OSF names StimulusSizeAndPosition.docx. 

-o0o-

REVIEWER #3

This paper seemed great in several ways: the question motivating the study is interesting and important, the results are clear and robust, and it's a well-written and well-organized manuscript. I also find this series of experiments to be a nice addition to the authors’ past work, and admire the depth of the research program as a whole. Along with my enthusiasm, I have a few concerns about the interpretation and discussion of the results, as well as some minor comments.

Thank you for the kind evaluation of our work.

Major comments

1. A crucial assumption underlying the conclusions is that the videos primed higher-level social interpretations; this was assessed via a final forced-choice question asking participants to select which stimuli were facing each other. Based on responses largely congruent with the primes, the authors conclude they have successfully manipulated “the representation of social agency in the direction of the object’s attention” (p.38). But the DV (i.e., participants’ ability to select the front as instructed) seems different from the conclusions (i.e., participants’ representations of social properties/joint attention). First of all, this forced-choice question is extremely susceptible to demand characteristics. For example, in Expts. 3/7 it would have been very surprising if participants had indicated the pointy/purple end of the seal as its front after seeing the speech bubble coming out of its round/yellow end. This is even more apparent in Expt. 5, where the instructions prompted participants to “pick which pair of seagulls are facing each other”; the fact the intended interpretation of the stimuli is re-iterated in the question makes me wonder whether participants were simply selecting what they had been taught was the correct option – regardless of their actual percepts. 

So I am on board when the authors label these primes as “semantic labels” (p. 26, 34), but not when they describe them as directly manipulating “the representation of social agency in the direction of the object’s attention” (p.38) or “high-level representations of social interactions” (p.38), or when equating these manipulations to the perception of a face or social interactions – which seems fundamentally different in nature.

We appreciate the reviewers concerns. Our initial hypothesis, and “hoped for” result, was to detect the effects of higher-level social representations in the visual search task. Unfortunately we did not detect these effects. Our approach has been to employ converging operations, where we attempted to detect the effects via a variety of techniques. These approaches have been established as effective. For example, the video action sequences have been used in many studies ranging from young children to clinical populations, and they always evoke potent experiences of animate interactive objects (see the video demos at osf.io/qxk8z). On the specific point concerning the measure of object identity biased by our techniques, first we would like to state that the orientation question results from Experiment 2 (no speech bubble and question does not refer to either side) are comparable to the results of Experiments 3 and 7 (target with speech bubble) and Experiment 5 (question with animal reference). This suggests that the question of front/face may not be as susceptible to demands as the reviewer suggests. 

In considerations of the reviewer’s strong feelings on the text on page 38, we have toned down and qualified the specified statements. And indeed, we do not rule out the possibility that as yet undiscovered techniques that are better than our approaches might demonstrate the role of high-level social representations in the search tasks.

2. Even assuming the primes worked as intended, I wonder if the relevant interpretations were active during the search task itself. Perceptual interpretations of ambiguous figures have been shown to rely on fixation patterns and covert attention (e.g. Peterson & Gibson, 1991, JEP:HPP; Toppino, 2003, P&P) – but both fixations and covert attention are key for successful visual search, and so top-down interpretations of ambiguous stimuli are directly in competition with search performance (which is especially high here). This is not a problem in itself, but it highlights another problem with generalizing from ambiguous stimuli to faces and people, since face perception doesn't require top-down control (or not even awareness, e.g. Stein et al., 2012, Cognition).

Indeed, we agree with the reviewer’s point. We actually suggest that the higher-level object identity processes (perhaps further eye-movements) may require some further processing, which is slower than the basic simple feature detection that triggers Posner-like orienting. Furthermore, we did not intend to generalise from our simple geometric and ambiguous figures directly to human faces and people generally. Rather we are investigating the influence of mechanisms processing lower-level features and implied ‘face/front’ by social priming. Note that we have been quite conservative in our General Discussion and suggest simply that higher-level social processes may have less influence at the earliest stages of processing than at later ones

3. Another assumption here is that default percepts of fronts reflect lower-level visual features; but I don't think this can be taken for granted. For example, the fact that seagulls facing each other are prioritized in Expt. 5 could be an effect of the interpretation of seagulls as seagulls (in both conditions, if the prime is inactive as per points #1 and #2) and it is in fact their perceived facing direction that is driving attention. 

Participants in Experiment 5 were assigned to either a seagull or a rabbit condition and all stimuli were referred to accordingly for each participant. There was no priority for seagulls. The experiment was counterbalanced as described in Methods.

The target was indeed driving attention with faster responses to seagull-inwards facing pairs regardless of prime condition being seagull or rabbit front. 

(Relatedly, the claim on p.7 that the stimuli “contain no intrinsic visual features that could be construed as a face” and have “no salient intrinsic face-like features” doesn’t seem right to me, since the seagulls/rabbits have a beak, ears, and eye.) 

We apologise. It was not clear this was actually referring to the stimuli in a different experiment, where the stimuli were presented in motion (teardrops and symmetrical shape). This has been amended. 

This also applies to Expt. 2, as some authors have suggested that arrows can in fact be socially meaningful (e.g. Kingstone et al., 2003) or suggest an agentic presence, especially when multiple stimuli point towards the same region of space (e.g. Gao et al., 2010, Psych Sci; Takahashi et al., 2013, Front Hum Neurosci; Colombatto et al., 2019, Perception). Since these results as described have potential implications for that literature, it would be great to see this discussed.

It remains debatable whether such stimuli have social properties. As we note, low-level shape properties imply/afford directional action based representations, such as arrows and the shape of diving gannets, and object motion patterns also afford directional cues even without social content (a falling leaf or moving cloud). Furthermore, Kingstone focusses more on arrows being socially meaningful rather than being social agents i.e. directing attention by feature rather than social interpretation. Gao’s work uses multiple moving arrows whose attention is towards a target square whereas Takahashi’s and Colombatto’s work uses multiple moving cones whose ‘attention’ is towards the participant observer. We feel these publications are quite different in focus than our current work which is exploring ascribed sociality to objects which can have feature cues in the opposite direction to the ‘social direction’. Further, it is difficult to project our findings onto less time-pressured situations as we point out in our discussion and as you indicate elsewhere in review. 

4. One of the reasons social binding was initially thought to be a social effect was that it vanished with inverted faces that are equivalent to upright faces in lower-level properties, but differ in the higher-level social properties (although related to my point #1, participants would still be able to indicate whether people are facing each other, even when they are inverted!). Those findings contradict the conclusion that lower-level properties only drive social binding – and since this work directly follows up on those experiments, it would be interesting to see the authors discuss this.

The reviewer is correct that related findings have previously been interpreted as social effects due to the sensitivity of the effects to inversion. This was thought to account for low-level features, such as symmetry, distance, and centre of mass, and therefore rule out lower-level mechanisms based on such features. However, more recently an explanation for such apparent social effects has been proposed based on directional cueing (Vestner, Gray, & Cook, 2020). This explanation suggests gaze cueing effects elicited by faces to be the cause for "social" interaction effects. These have been shown to be sensitive to stimulus inversion (Vestner, Gray, & Cook, 2021) while not requiring the stimulus itself to be perceived as social. It was later confirmed that these directional cueing effects are domain general and not specifically social, and can be extended to any number of non-social but directionally cueing objects (Vestner, Over, Gray, & Cook, 2021; Vestner, Over, Gray, Tipper, & Cook, 2021).

Minor comments

1. The main text for both Expts. 6 and 7 describes a ‘yellow’ vs. ‘purple’ manipulation (also depicted in Figs. 5, 14-15). But then Fig. 9 depicts ‘orange’ vs. ‘purple’ conditions, and the datafiles report ‘orange’ and ‘purple’ for Expt. 6, and ‘rabbit’ and ‘orange’ for Expt. 7 (in both the RT and MT files); could these be made consistent?

Thank you for pointing out this inconsistency. We have changed any ‘orange’ to ‘yellow’ in all documents. 

2. How were subjects assigned to the prime conditions – were they alternating? (This would be helpful to clarify as potentially related to block order assignment.)

Thank you for identifying this omission. Participants were indeed alternated (now described in Experiment 2 > Task blocks and social priming). 

3. How were the prime videos generated for the ‘round-face’ conditions? Were the stimuli simply mirrored/rotated? I ask because this seems to have produced some artifacts, e.g. the ‘round-faces’ overlap at the end of the videos and during the ball tossing game, which would be weird for agents, or the ‘round-face’ stimuli either toss the ball at a distance (rightmost), or overlap with the ball (leftmost).

Yes the teardrop stimuli were rotated 180° about their centre. We recognise that there are some gaps and overlaps of ball to characters during ‘throwing’ and between characters during close interaction but we don’t believe that this would be sufficient to break the illusion that the characters have intentions, opinions and feelings. Indeed, during initial testing of our stimuli observers always reported the animate nature of the objects in all versions of the videos (see video demos in OSF). 

4. The target orientation question for Exp. 2 is phrased in the instructions as “You need to pick which pair of seals are facing each other”, but I don’t think ‘seals’ was ever used elsewhere in Exp. 2. How could participants answer this question?

Well spotted! This was a copy/paste error. We have checked the experiment code and the script for the orientation question in Experiment 2 reads “You need to pick which pair of characters are facing each other”. We have the instructions document. 

5. This statement seems inaccurate/controversial: “higher-level mentalising processes are at play at later stages as suggested by […] behavioural studies showing that the effects of mentalising on gaze following are slower non-automatic”: the matter is still debated, but recent evidence showing the contrary should be cited, e.g. that mentalising such as perspective taking can occur rapidly and automatically (e.g. Ward et al., 2019, Curr Bio), and that percepts of mental states can influence lower-level processes such as gaze cueing, even quickly and automatically (e.g. Colombatto et al., 2020, PNAS).

Thank you for making us aware of this new work which we now cite in the manuscript.

6. For additional evidence that the pointy end of the teardrop is seen as its front, the authors may find helpful Chen & Scholl, 2018, PB&R – who used teardrop stimuli to elicit the same effects (on aesthetic preferences) as other types of fronts. (I wouldn't normally mention too many papers from our group in a review, but each one here seems warranted as potentially helpful or directly relevant to the discussion.)

Thank you for this suggestion. We assume the reviewer is referring to Experiment 3 of the 2014 paper ‘Seeing and liking: biased perception of ambiguous figures consistent with the “inward bias” in aesthetic preferences’. We have added this reference to our introduction.

---

## [Decision Letter · Decision Letter 1]

1 Sep 2021

PONE-D-21-06244R1

Rapid detection of social interactions is the result of domain general attentional processes

PLOS ONE

Dear Dr. Flavell,

Thank you for submitting your manuscript to PLOS ONE. After careful consideration, we feel that it has merit but does not fully meet PLOS ONE’s publication criteria as it currently stands.

Editor comment. Two of the previous referees commented again on your manuscript. As you can see from the reviews, they are positive about the new version and there are only some minor issues remaining. I would ask you to consider these points in a final revision of your manuscript, and I think further reviewer rounds will not be necessary. Therefore, we invite you to submit a revision of the manuscript that addresses the remaining points together with a cover letter that contains point-by-point replies.

We look forward to receiving your revised manuscript.

Kind regards,

Michael B. Steinborn, PhD

Section Editor, Cognitive Psychology

PLOS ONE

Journal Requirements:

Additional Editor Comments (if provided):

Reviewers' comments:

Reviewer's Responses to Questions

**Comments to the Author**

1. If the authors have adequately addressed your comments raised in a previous round of review and you feel that this manuscript is now acceptable for publication, you may indicate that here to bypass the “Comments to the Author” section, enter your conflict of interest statement in the “Confidential to Editor” section, and submit your "Accept" recommendation.

Reviewer #1: All comments have been addressed

Reviewer #3: (No Response)

2. Is the manuscript technically sound, and do the data support the conclusions?

Reviewer #1: Yes

Reviewer #3: Yes

3. Has the statistical analysis been performed appropriately and rigorously? 

Reviewer #1: Yes

Reviewer #3: Yes

4. Have the authors made all data underlying the findings in their manuscript fully available?

Reviewer #1: Yes

Reviewer #3: Yes

5. Is the manuscript presented in an intelligible fashion and written in standard English?

Reviewer #1: Yes

Reviewer #3: Yes

6. Review Comments to the Author

Reviewer #1: I feel that the authors have addressed all of my comments adequately and I strongly endorse publication.

Reviewer #3: The authors have addressed most of my earlier comments; I have a couple additional suggestions, but in general I believe with their revisions (along with their responses to other reviewers' comments), the manuscript is now in a better shape.

Regarding my point about labeling vs. perceiving social interactions, I see that the authors edited the example I provided, and I appreciated that the new introductory paragraph was worded accordingly. I disagree however that Experiment 2 rules out possible contributions of demand characteristics, since it looks like the instructions and question still mentioned “characters” that were “facing towards each other” – which is of course less apparent as a manipulation compared to speech bubbles, but could still plausibly be perceived as such.

In the previous round of review, I had discussed the results of Expt. 5, namely that seagulls facing each other are prioritized in visual search (and how it is consistent with a ‘social’ interpretation: if during the search task participants automatically perceive the stimuli as seagulls – regardless of which prime they had been exposed to – then the advantage for inward seagulls might reflect a prioritization of ‘facing’ vs. ‘non-facing’ dyads). The authors rule out this interpretation mentioning that (1) condition assignment was randomized between subjects, and (2) “there was no effect of seagulls”. For (1), I am not sure how random assignment might speak to the possibility that all participants might perceive the stimuli as seagulls during the search task, regardless of previous priming; and (2) also confused me, since my understanding is there was no effect of prime, but there was an effect of stimulus orientation for seagulls (faster for facing vs. non-facing seagulls, i.e. faster for non-facing vs. facing rabbits). I think this might just be a misunderstanding, but if not (i.e. if there was instead no effect of facing seagulls), the results section for Expt. 5 should be clarified.

I might have missed this, but I don’t think the stimulus generation method for the ‘round-face’ videos has been clarified in the manuscript?

I still see that the conditions of Expt. 7 labeled as ‘rabbit’ and ‘orange’ in the RT file, the MT file, and the data exclusion file.

The averages for Expt. 2 in the raw data file seem to be inconsistent with the figure (i.e. mean RT for inward point = 1014ms, inward round = 1003ms, but the depicted mean in Fig. 10 is higher for round). (As an aside, one would naturally associate pointed vs. rounded shapes with the point vs. round condition, but pointed vs. rounded shapes actually signal SOA in the figure, while point vs. round condition is signaled by color.)

The OSF stimuli folder for Exp. 6 contains priming videos and search targets – but wasn’t Exp. 6 simply a cueing task?

Apologies if I missed this in the manuscript, but it should be clarified that RT was computed as the time of spacebar release (i.e. without the reaching movement).

Both the main text and supplement state that in Expt. 1, 33 participants were tested and 26 remained following exclusions; but of the (presumably 7) excluded subjects, only 6 are mentioned (2 based on accuracy, and 4 based on RT).

7. PLOS authors have the option to publish the peer review history of their article (what does this mean?). If published, this will include your full peer review and any attached files.

Reviewer #1: **Yes: **Christina Breil

Reviewer #3: No

---

## [Author Response · Author response to Decision Letter 1]

21 Sep 2021

Please see 'Response to Reviewers.docx' for formmated version.

RESPONSE TO REVIEWERS

Dear Dr Steinborn,

Once more we’d like to thank you and the reviewers for the time and further comments concerning our paper “Rapid detection of social interactions is the result of domain general attentional processes”. We appreciate the opportunity to clarify and re-submit the work after. In the following we describe our response to the reviewer feedback. Reviewer comments are shown in black and our responses are shown in blue.

Yours sincerely,

Jonathan Flavell, Harriet Over, Tim Vestner, Richard Cook, Steven Tipper

-o0o-

REVIEWER #1

I feel that the authors have addressed all of my comments adequately and I strongly endorse publication.

Thank you for your time and for your recommendation.

REVIEWER #3

The authors have addressed most of my earlier comments; I have a couple additional suggestions, but in general I believe with their revisions (along with their responses to other reviewers' comments), the manuscript is now in a better shape.

Thank you for another thorough and considered review once again. 

Regarding my point about labeling vs. perceiving social interactions, I see that the authors edited the example I provided, and I appreciated that the new introductory paragraph was worded accordingly. I disagree however that Experiment 2 rules out possible contributions of demand characteristics, since it looks like the instructions and question still mentioned “characters” that were “facing towards each other” – which is of course less apparent as a manipulation compared to speech bubbles, but could still plausibly be perceived as such.

To clarify, in our previous Response to Reviewers we “…the question of front/face may not be as susceptible to demands…” rather than stating ruling them out entirely (which the present set of studies cannot do conclusively). 

Note that we have added some additional text to Experiment 2 > Results and Discussion.

In the previous round of review, I had discussed the results of Expt. 5, namely that seagulls facing each other are prioritized in visual search (and how it is consistent with a ‘social’ interpretation: if during the search task participants automatically perceive the stimuli as seagulls – regardless of which prime they had been exposed to – then the advantage for inward seagulls might reflect a prioritization of ‘facing’ vs. ‘non-facing’ dyads). The authors rule out this interpretation mentioning that (1) condition assignment was randomized between subjects, and (2) “there was no effect of seagulls”. For (1), I am not sure how random assignment might speak to the possibility that all participants might perceive the stimuli as seagulls during the search task, regardless of previous priming; and (2) also confused me, since my understanding is there was no effect of prime, but there was an effect of stimulus orientation for seagulls (faster for facing vs. non-facing seagulls, i.e. faster for non-facing vs. facing rabbits). I think this might just be a misunderstanding, but if not (i.e. if there was instead no effect of facing seagulls), the results section for Expt. 5 should be clarified.

Re. (1). Perhaps we had crossed wires. We thought that your description of seagulls as “prioritized in Expt. 5” was implying a methodological prioritisation. 

Re. (2). You are correct in that RTs were affected by seagull orientation but not by prime type. We cannot find the quote “there was no effect of seagulls” in our previous Manuscript or Response to Reviewers. Assuming you meant our words “There was no priority for seagulls” in Response to Reviewers, this is explained by the above point. 

We have edited the text in Experiment 5 > Results and Discussion and hope that this clarifies our own view on these data.

I might have missed this, but I don’t think the stimulus generation method for the ‘round-face’ videos has been clarified in the manuscript?

We did not realise that you were requesting an edit to the manuscript in your previous comment. We have now added the following to Experiment 2 > Method > Task blocks and social priming: “The pointed and rounded videos were identical apart from the orientation of the teardrop which was mirrored to create a pointed or rounded prime.”

I still see that the conditions of Expt. 7 labeled as ‘rabbit’ and ‘orange’ in the RT file, the MT file, and the data exclusion file.

Thankyou for identifying this error. These documents have now been corrected.

The averages for Expt. 2 in the raw data file seem to be inconsistent with the figure (i.e. mean RT for inward point = 1014ms, inward round = 1003ms, but the depicted mean in Fig. 10 is higher for round). (As an aside, one would naturally associate pointed vs. rounded shapes with the point vs. round condition, but pointed vs. rounded shapes actually signal SOA in the figure, while point vs. round condition is signaled by color.)

The figure has been corrected. The data are corrected but the figure labels were switched accidentally. We have corrected this and taken the opportunity to switch the icons in Figure 10 so that circles are for the rounded front condition and the triangles are for the pointed front condition.

The OSF stimuli folder for Exp. 6 contains priming videos and search targets – but wasn’t Exp. 6 simply a cueing task?

You are correct. There were no priming videos in Experiment 6 and these were uploaded in error. The six files have been removed from the OSF.

Apologies if I missed this in the manuscript, but it should be clarified that RT was computed as the time of spacebar release (i.e. without the reaching movement).

Thankyou for identifying this omission. We have added the following text to Experiment 2 > Method > Practice & trial composition: “Reaction times were measured from the moment the four simultaneously presented pairs appeared to the moment of space bar release. Movement time is measured from the moment of space bar release to the moment of screen contact.”

Both the main text and supplement state that in Expt. 1, 33 participants were tested and 26 remained following exclusions; but of the (presumably 7) excluded subjects, only 6 are mentioned (2 based on accuracy, and 4 based on RT).

We have double checked and it was the recruited size that was in error. It has been corrected to “Thirty-three two participants were tested…”. Thank you for identifying this mistake.

---

## [Editor Report · Decision Letter 2]

7 Oct 2021

Rapid detection of social interactions is the result of domain general attentional processes

PONE-D-21-06244R2

Dear Dr. Flavell,

We’re pleased to inform you that your manuscript has been judged scientifically suitable for publication and will be formally accepted for publication once it meets all outstanding technical requirements.

Kind regards,

Michael B. Steinborn, PhD

Section Editor

PLOS ONE
---

## [Editor Report · Acceptance letter]

7 Jan 2022

PONE-D-21-06244R2 

Rapid detection of social interactions is the result of domain general attentional processes 

Dear Dr. Flavell:

I'm pleased to inform you that your manuscript has been deemed suitable for publication in PLOS ONE. Congratulations! Your manuscript is now with our production department. 

Kind regards, 

on behalf of

Dr. Michael B. Steinborn 

Section Editor

PLOS ONE